# Maximum Likelihood Reinforcement Learning

**Fahim Tajwar** [* 1]  **Guanning Zeng** [* 1]  **Yueer Zhou** [2]  **Yuda Song** [1]  **Daman Arora** [1]  **Yiding Jiang** [1]  **Jeff Schneider** [1]
**Ruslan Salakhutdinov** [1]  **Haiwen Feng** [3 4]  **Andrea Zanette** [1]

## Abstract

Reinforcement learning (RL) is the method of choice for training models in setups where the objective function can only be evaluated by sampling from the model. Our key observation is that when the feedback is terminal and binary, models implicitly induce a likelihood over correct rollouts. Maximum likelihood would be the natural framework in such settings, but RL is used instead as a workaround to the non-differentiability. We prove that the standard, expected-reward RL formulation is only a first-order approximation of the likelihood. To remedy this mismatch, we introduce **Maximum Likelihood Reinforcement Learning (MaxRL)**, a compute-indexed family of sample-based objectives that interpolate between expected-reward RL and maximum likelihood as sampling compute is scaled. The resulting objective is a one-line change to standard RL implementations. MAXRL Pareto-dominates existing methods in all tested models and tasks, achieves up to $20\times$ gains in test-time scaling efficiency over GRPO, and scales more favorably with additional training data and compute.

## 1. Introduction

Maximum likelihood (ML) and reinforcement learning (RL) are two highly successful optimization paradigms that significantly shaped the landscape of modern machine learning. Maximum likelihood training is a foundational principle behind modern generative and predictive models (Bishop, 2006; Murphy, 2012); in fully differentiable settings, optimizing log-likelihood objectives has reliably translated increases in model capacity, data, and compute into consis-

tent performance improvements (Krizhevsky et al., 2012; Radford et al., 2018). Reinforcement learning, by contrast, originated in optimal control and sequential decision-making (Bertsekas, 1995; Sutton et al., 1998), where learning proceeds through interaction with an environment and the objective is to maximize expected return. The generality of this formulation enables reinforcement learning to address problems involving *non-differentiable* intermediate sampling, and has led to models capable of superhuman performance in complex domains (Mnih et al., 2015; Silver et al., 2016; OpenAI et al., 2019; Vinyals et al., 2019).

Many modern learning problems are typically addressed with reinforcement learning even if they define an implicit likelihood over successes. Examples include navigation (Thrun et al., 2005; Anderson et al., 2018), program synthesis (Chen et al., 2018; Bunel et al., 2018), structured prediction (Smith, 2011; Mensch & Blondel, 2018), and multi-step reasoning in large language models (Wei et al., 2022; Guo et al., 2025). In these tasks, success is determined by an external verifier only after a stochastic generation process, yielding a binary outcome. From an end-to-end perspective, the model induces a probability of success for each input, defining an implicit likelihood over correctness. Maximizing this likelihood would be the principled approach, but non-differentiable intermediate sampling precludes direct optimization. Reinforcement learning is used instead, not because it offers a better objective, but as a workaround to this non-differentiability.

Suppose that both maximum likelihood and reinforcement learning could be applied to a given task, regardless of differentiability. Then the two objectives induce markedly different optimization behavior, as we shall explain below. To make this distinction precise, we compare the population-level objectives implied by each framework. Let $p_\theta(x)$ denote the probability that a model with parameters $\theta$ produces a correct output for input $x$. Moreover, let $\rho$ be the distribution from which the inputs are sampled. The corresponding gradients of reinforcement learning, $\nabla_\theta J_{\mathrm{RL}}$, and of maximum likelihood, $\nabla_\theta J_{\mathrm{ML}}$, take the form

$$\nabla_\theta J_{\mathrm{RL}} = \mathbb{E}_{x\sim\rho}[\nabla_\theta p_\theta(x)]$$

$$\nabla_\theta J_{\mathrm{ML}} = \mathbb{E}_{x\sim\rho}[\nabla_\theta \log p_\theta(x)] = \mathbb{E}_{x\sim\rho}\left[\frac{1}{p_\theta(x)}\nabla_\theta p_\theta(x)\right]$$

---
[*]Equal contribution  [1]Carnegie Mellon University  [2]Stanford University  [3]UC Berkeley  [4]Impossible, Inc.  Correspondence to: Fahim Tajwar <ftajwar@andrew.cmu.edu>, Guanning Zeng <guanninz@andrew.cmu.edu>, Andrea Zanette <azanette@andrew.cmu.edu>.

*Proceedings of the 43$^{rd}$ International Conference on Machine Learning*, Seoul, South Korea. PMLR 306, 2026. Copyright 2026 by the author(s).

The inverse-probability reweighting induced by maximum likelihood places greater emphasis on hard, low-success inputs, leading to very different optimization dynamics, as we empirically demonstrate in this paper (cf. Section 6).

When viewed end-to-end, maximum likelihood emerges as the principled objective, for the very same reasons it is the method of choice in differentiable supervised learning with binary correctness. In non-differentiable problems, however, this objective is difficult to optimize directly because correctness is observed only after a non-differentiable stochastic generation process, and the success probability $p_\theta(x)$ may be small. This computational challenge motivates a framework that leverages additional sampling compute to more faithfully approximate likelihood-based training. We call this framework **Maximum Likelihood Reinforcement Learning (MaxRL)**. At a high level, MAXRL bridges standard reinforcement learning and exact maximum likelihood by progressively incorporating higher-order correctness information as more sampling compute is utilized, ultimately recovering likelihood optimization in the infinite-compute limit. We discuss the relationship between MAXRL and recent work in this direction (Xiong et al., 2025b; Davis & Recht, 2025) in Section 7. Our contributions are threefold:

1. We formalize correctness-based reinforcement learning as a latent-generation maximum likelihood problem and show that standard reinforcement learning optimizes only the first-order approximation of the maximum likelihood objective (cf. Section 3).

2. We introduce a compute-indexed family of objectives that interpolates between expected reward and exact maximum likelihood via a Maclaurin expansion in pass@k events (cf. Section 3).

3. We analyze a simple on-policy estimator whose expected gradient exactly matches the compute-indexed approximation of the likelihood objective, implying that increased sampling improves the optimized objective itself rather than merely reducing variance.

Empirically, MAXRL Pareto-dominates standard RL objectives (RLOO (Ahmadian et al., 2024), GRPO (Shao et al., 2024)) on all settings that we tested (cf. Section 6). Specifically, MAXRL shows better scaling trends when additional compute and data are available, and on mathematical reasoning tasks, it achieves up to $20\times$ test-time scaling efficiency gains over GRPO with a perfect verifier.

## 2. Preliminaries

We focus on correctness-based problems which define a *binary* success or failure outcome for each input. Formally, let $x \sim \rho$ be an input drawn from a distribution $\rho$, and let $y^*(x)$ denote the correct label or answer for $x$. Let the learner be parameterized by $\theta$, inducing a distribution $p_\theta(\cdot \mid x)$ over outputs for each input $x$. Equality between outputs is defined up to a task-dependent equivalence relation, so that $y = y^*(x)$ denotes semantic correctness rather than exact output equality. For example, in mathematical reasoning, $x$ is the prompt and $y$ is the final solution produced by the model. We use natural logarithms unless stated otherwise.

**Latent generation models.** In many settings, the model generates a latent variable $z \in \mathcal{Z}$ (a trajectory, e.g. a chain of thought) according to a discrete conditional distribution $m_\theta(z \mid x)$ before producing the final answer. The final output $y$ is then obtained via a deterministic decoding function $y = f(z)$, such as parsing a generated program or extracting a boxed answer from a chain of thought. Correctness is evaluated only on the decoded output, i.e., $z$ is successful if $f(z) = y^*(x)$. Decoding pushes $m_\theta$ forward to the output distribution $p_\theta(\cdot \mid x)$ from above,

$$p_\theta(y \mid x) = \sum_{z \in \mathcal{Z}} m_\theta(z \mid x) \, \mathbb{I}\{f(z) = y\}. \qquad (1)$$

Throughout the paper, expectations with respect to model outputs should be understood as expectations over latent samples $z \sim m_\theta(\cdot \mid x)$ followed by deterministic decoding.

**Pass rate.** We define the *pass rate* as the probability that the model produces the correct answer for a fixed input $x$:

$$p_\theta^{\text{pass}}(x) := p_\theta(y^*(x) \mid x) = \mathbb{E}_{y \sim p_\theta(\cdot \mid x)}[\mathbb{I}\{y = y^*(x)\}].$$

Similarly, let $y_1, \ldots, y_k \overset{\text{i.i.d.}}{\sim} p_\theta(\cdot \mid x)$. We define pass@$k$ as the probability of at least one correct sample:

$$\text{pass@}k(x) := \mathbb{P}(\exists\, i \in [k] \text{ s.t. } y_i = y^*(x)).$$

**Maximum likelihood (ML).** Maximum likelihood selects parameters that maximize the *log-likelihood* of the observed data under the model. In our setting, the probability of observing correctness is $p_\theta(y^*(x) \mid x)$. Maximizing the corresponding log-likelihood therefore yields the objective

$$J_{\text{ML}}(\theta) := \mathbb{E}_{x \sim \rho}[\log p_\theta(y^*(x) \mid x)], \qquad (2)$$

which is directly analogous to cross-entropy training in differentiable supervised learning.

**Reinforcement learning (RL).** Typically, reinforcement learning is applied to correctness-based tasks by defining a binary reward function $r(x, y) = \mathbb{I}\{y = y^*(x)\}$ indicating correctness, yielding the RL objective:

$$J_{\text{RL}}(\theta) := \mathbb{E}_{x \sim \rho}[\mathbb{E}_{z \sim m_\theta(\cdot \mid x)}[r(x, z)]] = \mathbb{E}_{x \sim \rho}[p_\theta^{\text{pass}}(x)]. \qquad (3)$$

In short, RL maximizes the expectation over *pass rate* while ML maximizes the expectation over its *logarithm*.

# 3. Maximum Likelihood Reinforcement Learning (MAXRL)

In this section, we show that reinforcement learning on expected reward optimizes only a first-order approximation of the ML objective. Specifically, the maximum likelihood objective admits a population-level expansion in terms of pass@k events, with standard RL optimizing only the first-order term. This suggests a compute-indexed family of objectives that incorporate higher-order terms, converging to ML as more compute is allocated.

## 3.1. Maclaurin Expansion of Maximum Likelihood

For simplicity, let us consider a single task $x$ in the development to follow, as the final objective and gradients can be obtained by taking an expectation over $x \sim \rho$. Moreover, we write $p := p_\theta^{\text{pass}}(x)$ to simplify our notation. The maximum likelihood objective admits the *Maclaurin expansion* in terms of failure events:

$$J_{\text{ML}}(x) = \log p = -\sum_{k=1}^{\infty} \frac{(1-p)^k}{k} = -\sum_{k=1}^{\infty} \frac{\text{fail@}k(x)}{k},$$
(4)

where $\text{fail@}k(x) = 1 - \text{pass@}k(x)$ denotes the probability that all $k$ i.i.d. samples from the model fail. Differentiating Equation (4) yields the population-level gradient identity

$$\boxed{\nabla_\theta J_{\text{ML}}(x) = \sum_{k=1}^{\infty} \frac{1}{k} \nabla_\theta \text{pass@}k(x).}$$
(5)

Thus, maximum likelihood optimizes an infinite harmonic mixture of $\text{pass@}k$ gradients, with higher-order terms encoding rare success which are critical when $p$ is small. In contrast, the classical reinforcement learning approach is to optimize only the expected pass@1 objective (Koenig & Simmons, 1993; Silver et al., 2016; Vecerik et al., 2018; Guo et al., 2025):

$$\nabla_\theta J_{\text{RL}}(x) = \nabla_\theta \text{pass@}1(x),$$

corresponding to retaining solely the leading term of Equation (5). From this observation, we can claim:

*Reinforcement learning optimizes a first-order approximation of the maximum likelihood objective.*

## 3.2. MAXRL Objective Function

The maximum likelihood gradient in Equation (5) is difficult to estimate with a fixed number of samples. In particular, estimating pass@k gradients for large $k$ requires an increasing number of samples, especially when the pass rate $p$ is small. This finite-sample difficulty is precisely what motivates Maximum Likelihood Reinforcement Learning. We define MAXRL as the *class of reinforcement-learning methods that explicitly target the maximum likelihood objective* rather than the pass rate, while remaining implementable under fixed sampling and non-differentiable generation. We consider a principled way to do so below.

Consider approximating the maximum likelihood objective by truncating the Maclaurin expansion (Equation (5)) to a fixed finite order and then estimating such an objective instead. For a truncation level $T \in \mathbb{N}$, we define the truncated maximum likelihood objective for a fixed input $x$ as:

$$J_{\text{MAXRL}}^{(T)}(x) := -\sum_{k=1}^{T} \frac{(1-p)^k}{k}.$$
(6)

Differentiating Equation (6) yields the truncated population gradient:

$$\nabla_\theta J_{\text{MAXRL}}^{(T)}(x) = \sum_{k=1}^{T} \frac{1}{k} \nabla_\theta \text{pass@}k(x).$$
(7)

This defines a family of objectives: **$T = 1$ recovers reinforcement learning, $T \to \infty$ recovers maximum likelihood**, and intermediate $T$ values interpolate between them. Thus, the truncation level $T$ directly controls the order of correctness events that contribute to learning. As we will soon see, it becomes viable to estimate higher-order $\nabla_\theta J_{\text{MAXRL}}^{(T)}(x)$ as more compute is expended in terms of rollouts. In other words:

*MAXRL provides a principled framework for trading additional compute for higher-fidelity approximations to the maximum likelihood objective.*

The remaining question is whether these truncated objectives admit simple, unbiased estimators under a fixed sampling budget, a question that we answer affirmatively in the next section.

# 4. Gradient Estimators for MAXRL

Equation (7) already provides a viable approach for constructing an unbiased estimator: approximate *each* term in the finite series using a pass@k gradient estimator, as provided in recent work (Walder & Karkhanis, 2025; Chen et al., 2025b). Under this strategy, any improvement in pass@k estimators directly translates into improved estimators for the truncated maximum likelihood objective that we provided in Equation (7).

In this work, we take an alternate approach. The key insight is that the maximum likelihood gradient can be expressed as an expectation under the *success-conditioned* distribution, as established by the following theorem (Davis & Recht (2025) also recently made a similar observation).

**Theorem 4.1** (Conditional Form of the Maximum Likelihood Gradient). *The gradient of the maximum likelihood objective admits the following conditional expectation representation:*

$$\nabla_\theta J_{\mathrm{ML}}(x) = \mathbb{E}[\nabla_\theta \log m_\theta(z \mid x) \mid f(z) = y^*(x)]. \quad (8)$$

We provide the proof of this theorem in Appendix C. The theorem establishes that the maximum likelihood gradient is the average gradient from successful trajectories only. This interpretation naturally leads to a concrete gradient estimator by replacing the expectation with sample averages over successful rollouts.

### 4.1. Empirical Gradient Estimator

*Table 1.* Comparison of the REINFORCE estimator ($\nabla_\theta J_{\mathrm{RL}}(x)$) and the conditional estimator ($\nabla_\theta J_{\mathrm{MAXRL}}^{(T)}(x)$). Although they differ only in normalization ($N$ vs. $K$), they are unbiased for fundamentally different objectives.

|            | **REINFORCE**                  | **MAXRL**                                         |
| ---------- | ------------------------------ | ------------------------------------------------- |
| Estimator  | $\frac{1}{N}\sum_{i=1}^{N} r_i S_i$ | $\frac{1}{K}\sum_{i=1}^{N} r_i S_i$          |
| Unbiased for | $\nabla_\theta \, \mathrm{pass@1}(x)$ | $\sum_{k=1}^{N} \frac{1}{k} \nabla_\theta \mathrm{pass@}k(x)$ |

Theorem 4.1 suggests drawing samples from the success-conditioned policy. Recent literature has proposed rejection fine-tuning (Touvron et al., 2023; Yuan et al., 2023; Dong et al., 2023; Xiong et al., 2025a; Davis & Recht, 2025) and adaptive sampling (Xiong et al., 2025b) as mechanisms to sample from this conditional distribution. However, doing so is computationally demanding when the pass rate is small or requires a more complex implementation regarding adaptive sampling. Instead, we adopt a simpler approach: we sample from the unconditional policy $m_\theta(\cdot \mid x)$ and then *average over only the successful trajectories*. Fix an input $x$ and draw $N$ latent trajectories $z_1, \ldots, z_N \sim m_\theta(\cdot \mid x)$. Let $r_i := \mathbb{I}\{f(z_i) = y^*(x)\}$ indicates success-based reward, $S_i := \nabla_\theta \log m_\theta(z_i \mid x)$ denote the score function, and $K := \sum_{i=1}^{N} r_i$ be the number of successful samples. We average score functions *only over successful trajectories* and obtain the following REINFORCE-style estimator:

$$\widehat{g}_N(x) := \begin{cases} \dfrac{1}{K} \sum_{i=1}^{N} r_i S_i, & K \geq 1, \\ 0, & K = 0. \end{cases} \quad (9)$$

The estimator constructed in this way is such that some inputs may receive zero gradient if there are no successes within $N$ samples, making the resulting estimator no longer unbiased with respect to the infinite series expansion of the true ML objective from Equation (5). We show that this estimator is however unbiased for the gradient of the truncated maximum likelihood objective in Equation (7), $\nabla_\theta J_{\mathrm{MAXRL}}^{(T)}(x)$, with truncation level $T = N$:

**Theorem 4.2** (Estimator–objective equivalence). *The estimator $\widehat{g}_N(x)$ is an unbiased estimator for the MAXRL gradient of order $T = N$, i.e.,*

$$\mathbb{E}[\widehat{g}_N(x)] = \nabla_\theta J_{\mathrm{MAXRL}}^{(N)}(x).$$

We present the proof of this result in Appendix C. Theorem 4.2 reveals an elegant alignment between the estimator in Equation (9) and the gradient of the truncated Maclaurin expansion in Equation (7). It is worth highlighting the most important property of the estimator:

*Increasing compute as rollouts $N$ leads to a better approximation of the maximum likelihood gradient.*

Table 1 compares our estimator with the REINFORCE estimator, whose expected value underlies most RL algorithms. At the estimator level, the difference is simple: both average score functions over sampled trajectories, but REINFORCE normalizes by total samples $N$ while MAXRL normalizes by successful samples $K$. This difference in normalization determines the objective each estimator is unbiased for.

Consequently, increasing the number of samples, $N$, for the two estimators has different effects: REINFORCE[1] reduces variance of a fixed objective (pass@1), while MAXRL increases the approximation order to maximum likelihood. Additional compute thus improves the *objective itself* for MAXRL, not just estimation quality.

### 4.2. Variance Reduction via Control Variates

Like REINFORCE, the estimator in Equation (9) can exhibit high variance when the number of successful samples, $K$, is small. Policy-gradient baselines are typically introduced to reduce variance without changing the expected gradient (Sutton, 1988). However, standard arguments for policy-gradient baselines are not directly applicable in this setting, as the estimator normalizes by the random variable $K$ which depends on all samples, making it correlated with the observed rollouts.

We instead proceed from first principles and use a simple zero-mean control variate, the unconditional average score:

$$V_N := \frac{1}{N} \sum_{i=1}^{N} \nabla_\theta \log m_\theta(z_i \mid x).$$

---

[1]Modern policy-gradient methods such as PPO (Schulman et al., 2017) introduce additional mechanisms (importance weight truncation via clipping) that trade bias for robustness. In the fully on-policy setting, these reduce to REINFORCE, our canonical baseline. GRPO (Shao et al., 2024) is a notable exception due to its division by standard deviation in the advantage calculation, which we discuss further in Section 5.

This satisfies $\mathbb{E}[V_N] = 0$. Subtracting $V_N$ preserves unbiasedness while reducing variance in practice:

$$\widetilde{g}_N(x) = \frac{1}{K}\sum_{i=1}^{N} r_i S_i - \frac{1}{N}\sum_{i=1}^{N} S_i = \sum_{i=1}^{N}\left(\frac{r_i}{K} - \frac{1}{N}\right) S_i \tag{10}$$

with the convention that the first term, $\left(\sum_{i=1}^{N} r_i S_i\right)/K$, is zero when $K = 0$.

### 4.3. On-Policy Implementation

Finally, we present a simple *on-policy* implementation of MAXRL that differs from standard REINFORCE-style policy gradient methods by a *single-line* modification to the advantage calculation. We adopt the variance-reduced formulation from Equation (10), and drop both terms when $K = 0$, consistent with standard policy-gradient practice (Yu et al., 2025; Nie et al., 2026) in which no gradient is computed for tasks with no successful rollouts: this choice is simpler and performs better empirically. Concretely, the advantage is normalized by the per-task mean reward, rather than left unnormalized as in RLOO (Ahmadian et al., 2024) or normalized by the reward standard deviation as in GRPO (Shao et al., 2024). The modified line is highlighted in blue.

---

**Algorithm 1 On-Policy Implementation of MAXRL**

---

1: **Input:** Batch of inputs $B$, number of rollouts $N$, latent policy $m_\theta$
2: **for** each input $x \in B$ **do**
3:     Sample $N$ rollouts $z_1, \ldots, z_N \overset{\text{i.i.d.}}{\sim} m_\theta(\cdot \mid x)$
4:     Compute rewards $r_j = \mathbb{I}\{f(z_j) = y^*(x)\}$
5:     Compute mean reward $\widehat{r}(x) = \frac{1}{N}\sum_{j=1}^{N} r_j$
6:     Set

$$\widetilde{g}(x) = \begin{cases} \dfrac{1}{N\,\widehat{r}(x)}\sum_{j=1}^{N}(r_j - \widehat{r}(x))S_j, & \widehat{r}(x) > 0, \\ 0, & \text{otherwise.} \end{cases}$$

7: **end for**
8: Set $\widetilde{g} = \frac{1}{|B|}\sum_{x \in B}\widetilde{g}(x)$
9: **return** $\widetilde{g}$

---

We direct the reader to Section M.2 for a study on the effect of the control variate baseline on gradient variance: Figure 28 shows that despite having no theoretical guaranties, baseline subtraction generally reduces gradient variance in practice, especially when the rollout count is low. Finally, it should be noted that the practical gradient estimator in Algorithm 1 is an unbiased estimator for the MAXRL gradient of order $T = N - 1$, as opposed to order $T = N$ from Theorem 4.2. This is due to the lack of baseline subtraction for tasks without any positive rollouts, which changes the expected gradient. See Section D for a detailed analysis.

## 5. A Unifying Weight-Function View

Maximum likelihood, MAXRL, classical reinforcement learning, and GRPO all admit population-level gradients of the following form:

$$\nabla_\theta J = \mathbb{E}_{x\sim\rho}[w(p_\theta(x))\ \nabla_\theta p_\theta(x)] \tag{11}$$

where $p_\theta(x) = p_\theta^{\text{pass}}(x)$ and $w(p)$ is a scalar weight that depends only on the pass rate. The function $w(p)$ determines how learning signal is allocated across inputs of varying difficulty and fully characterizes the differences between these objectives at the population level. Figure 1 plots the resulting weight functions for each method; derivations are provided in Appendix E. The key distinction among objectives is *how strongly they emphasize hard, low-pass-rate inputs*. As $T$ increases, MAXRL uniquely approaches maximum likelihood weighting in the low-pass-rate regime.

*Table 2.* Population-level weighting functions $w(p)$.

| | RL | GRPO | MAXRL ($T$) | ML |
|---|---|---|---|---|
| $w(p)$ | $1$ | $\dfrac{1}{\sqrt{p(1-p)}}$ | $\dfrac{1-(1-p)^T}{p}$ | $\dfrac{1}{p}$ |

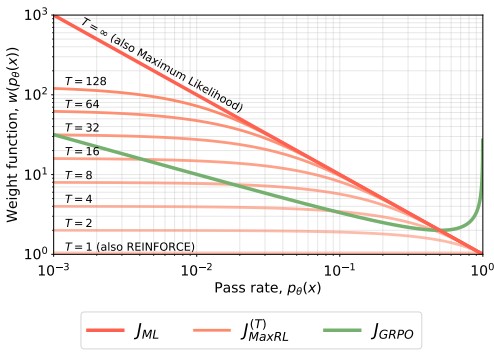

*Figure 1.* Population-level weighting functions $w(p)$ as a function of pass rate $p$. Truncated objectives, $J_{\text{MAXRL}}^{(T)}$, interpolate between REINFORCE and maximum likelihood as $T$ increases.

This weight perspective also provides a useful reinterpretation of GRPO (Shao et al., 2024). Although GRPO is heuristically motivated by Z-normalization using the empirical standard deviation, such normalization induces a fundamentally different population-level objective from REINFORCE, a conclusion also reached by recent work (Davis & Recht, 2025; Liu et al., 2025b; Xiong et al., 2025b; Yang et al., 2026a; Thrampoulidis et al., 2026). Table 2 summarizes the population-level weighting functions; relative to standard expected-reward optimization, GRPO upweights low-pass-rate inputs approximately as $1/\sqrt{p}$ when $p$ is small, placing it between classical reinforcement learning and maximum likelihood. However, increasing compute via additional sampling under GRPO does not yield a better approximation to the maximum likelihood objective, as the induced

population loss is fundamentally distinct. Moreover, as shown in Figure 1, the GRPO weighting function *inverts* for sufficiently large pass rates, increasing as $p \to 1$, unlike likelihood-based objectives. Consequently, GRPO assigns increased weight to very easy inputs when they are present, in contrast to the other formulations.[2]

# 6. Experiments

We now turn to the empirical evaluation of MAXRL. We begin in Section 6.1 with a controlled setting where exact maximum likelihood optimization is possible, allowing direct comparison with MAXRL as compute increases. We then study non-differentiable correctness-based tasks in two regimes: (i) an *effectively infinite-data* setting with large number of novel tasks (Section 6.2), and (ii) a *data-scarce* setting with a fixed training dataset where we can scale compute nonetheless by training for many epochs over the same dataset (Section 6.3). Finally, in Section 6.4, we train and evaluate billion-parameter reasoning models on mathematical problem-solving, testing whether the benefits of MAXRL extend to larger-scale LLM training.

Because we compare training objectives rather than algorithms, all methods are trained *on-policy*. We compare against RLOO (Ahmadian et al., 2024) and GRPO (Shao et al., 2024) as primary baselines.

## 6.1. Comparisons with Exact Likelihood

As a first step, we evaluate how closely MAXRL approximates *exact* maximum likelihood in a setting where the latter can be implemented exactly. We compare three objectives: (i) reinforcement learning on expected reward, (ii) MAXRL (a fixed sample-based objective), and (iii) exact maximum likelihood training. We consider a standard image classification task, where the exact ML objective corresponds to minimizing cross-entropy. The RL reward is defined as 1 if the predicted class matches the ground-truth label and 0 otherwise. We instantiate this comparison on the ImageNet (Deng et al., 2009) dataset using a ResNet-50 (He et al., 2016) based classifier trained under each objective; full experimental details are provided in Appendix H. Figure 2 summarizes the results: REINFORCE (with a standard baseline) fails to achieve meaningful improvements even with very high per-input sampling budget, whereas exact maximum likelihood training yields steady gains on both average performance (Pass@1) and coverage (Pass@k).

In contrast, MAXRL is trained on the same samples and observes the same sparse set of successful trajectories as

---

[2]We conjecture that this inversion may contribute to distribution sharpening (Pass@k degradation) (Yue et al., 2025; Wu et al., 2026) when datasets contain a substantial fraction of overly easy inputs, and leave a detailed analysis to future work.

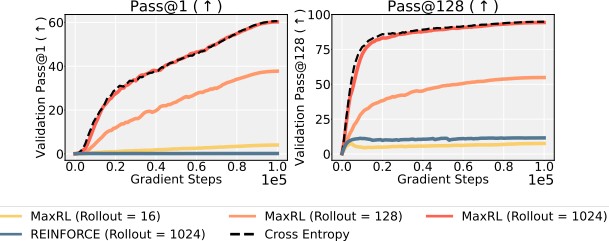

*Figure 2.* (**ImageNet**) Comparison of training dynamics under exact maximum likelihood, MAXRL, and REINFORCE in a controlled image classification setting. With sufficient rollouts, MAXRL closely matches cross-entropy training, while REINFORCE fails to make progress from low initial pass rates.

REINFORCE, but makes more effective use of this limited learning signal through likelihood-inspired reweighting. As the compute increases by means of higher rollout counts, MAXRL *improves consistently and closely tracks exact maximum likelihood.* We also analyze the gradient norm resulting from different objectives in Figure 8: MAXRL and cross-entropy concentrate learning signal on harder tasks and are characteristically similar given sufficient compute for MAXRL, whereas GRPO and REINFORCE exhibit very different behavior. For additional experiments such as comparison to GRPO, we refer the reader to Appendix H.

> **Takeaway 1**
> When direct maximum-likelihood optimization is available, MAXRL converges to it as sampling compute increases.

## 6.2. Infinite Data Regime

Next, we study MAXRL in non-differentiable settings. For the first experiment, we study how MAXRL behaves in data-rich domains. To simulate training with infinite data, we construct a procedurally generated maze-navigation environment with 1 million unique $17 \times 17$ mazes for training where multiple valid solution paths might exist for a given task. We reserve a held-out set of 256 mazes for evaluation and apply a brief supervised pretraining phase to ensure a non-zero initial pass rate. The complete details of the task are provided in the Appendix I.

We train a lightweight transformer model (Vaswani et al., 2017) with approximately 3M parameters and simulate extended training by running 9,000 RL steps with a batch size of 256. We report performance after 9,000 steps in Figure 3 as a function of training compute, implemented as different number of rollouts per prompt, from 4 to 128, while keeping other hyperparameters the same. Notice that this is a substantial amount of compute for our particular model size.

All three objectives (RLOO, GRPO, and MAXRL) improve upon the base model, but the magnitude of improvement differs markedly across methods. Even at the highest compute budget (128 rollouts per prompt), RLOO fails to match the performance achieved by MAXRL at the lowest budget (4

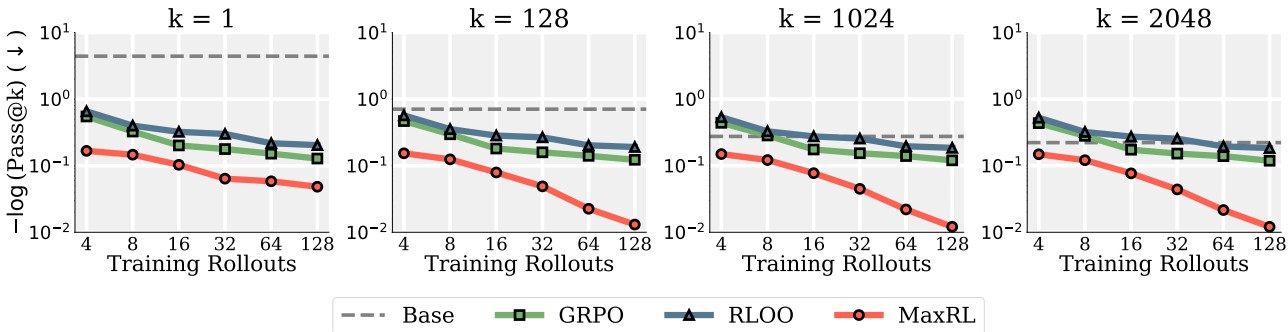

*Figure 3.* (**Maze**) Motivated by Schaeffer et al. (2025), we record $-\log(\text{pass@}k)$ (lower is better) as a function of training rollouts for different objectives. We see that across all different inference rollout budgets ($k$), MAXRL exhibits better scaling compared to GRPO and RLOO as we increase number of training rollouts.

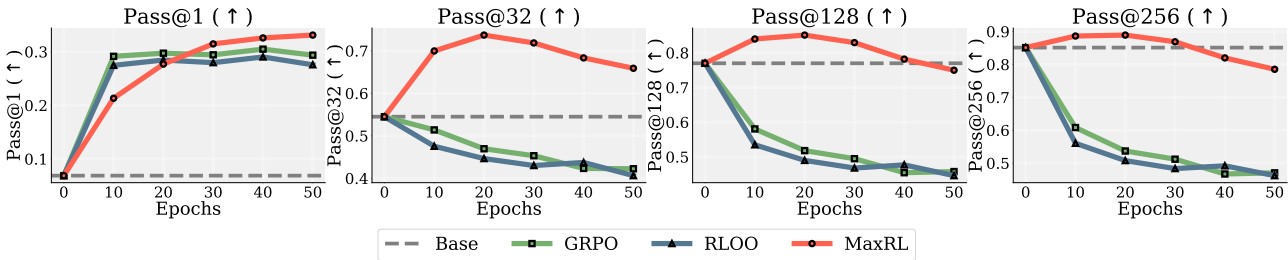

*Figure 4.* (**GSM8K**) Training dynamics on GSM8K with a fixed dataset and increasing training compute in terms of RL steps. MAXRL shows slower initial gains but ultimately achieves higher performance with substantially less pass@k degradation.

rollouts) across all pass@k metrics. GRPO-trained models at the highest compute similarly trail MAXRL trained with only 16 rollouts per prompt at every pass@k evaluation. These results highlight that MAXRL *scales with compute* far more effectively than competing frameworks when large amounts of unique training data are available. Moreover, there is almost no gap in pass@$k$ and pass@1 for GRPO and RLOO *despite no repeated data*, showing that these algorithms have lost the ability to improve further, whereas MAXRL has a substantial gap and potentially can improve more given additional compute. Comparisons with additional baselines are provided in Table 3 and Appendix L.

> **Takeaway 2**
>
> In a data-rich training regime, MAXRL scales more favorably with additional compute compared to existing methods.

### 6.3. Data-Scarce Regime

We next consider a data-scarce regime where models are trained for many epochs on a fixed dataset until peak performance is reached, with the goal of identifying which method extracts the highest attainable performance. Unlike the infinite-data setting in Section 6.2, longer training in this regime does not necessarily translate into improved performance due to the increased risk of overfitting. Specifically, we train a SmolLM2-360M-Instruct model (Allal et al., 2025) on GSM8K (Cobbe et al., 2021) for up to 50 epochs. Training dynamics is reported in Figure 4, with additional experimental details provided in Appendix J.

All methods improve on the base model in terms of pass@1 performance; however, only MAXRL consistently exceeds the base model in pass@k metrics. In contrast, RLOO and GRPO exhibit massive pass@k degradation with extended training, mirroring the behavior observed by Yue et al. (2025) and exacerbated here by prolonged training on a fixed dataset. RLOO and GRPO reach their peak pass@1 performance faster than MAXRL (around 10 epochs), but MAXRL overtakes competing methods at approximately 30 epochs and continues to increase through the end of training, reaching a higher peak. At the same time, pass@k remains substantially healthier for MAXRL and exceeds the base model for a large portion of training. This behavior suggests MAXRL *is more resistant to overfitting*, particularly with regard to output diversity. Additional comparisons with baseline methods are reported in Table 5.

> **Takeaway 3**
>
> In a data-scarce regime, MAXRL can sustain improvement over a large number of epochs, demonstrating less pass@k degradation (overfitting) and converging to a higher average performance.

### 6.4. Large Reasoning Model Training

We next demonstrate that the benefits of MAXRL extend to larger-scale LLM reasoning training. We train Qwen3-1.7B-Base and Qwen3-4B-Base models on POLARIS-53K (An et al., 2025), a dataset of approximately 50,000 mathematical reasoning prompts, using 256 prompts per batch,

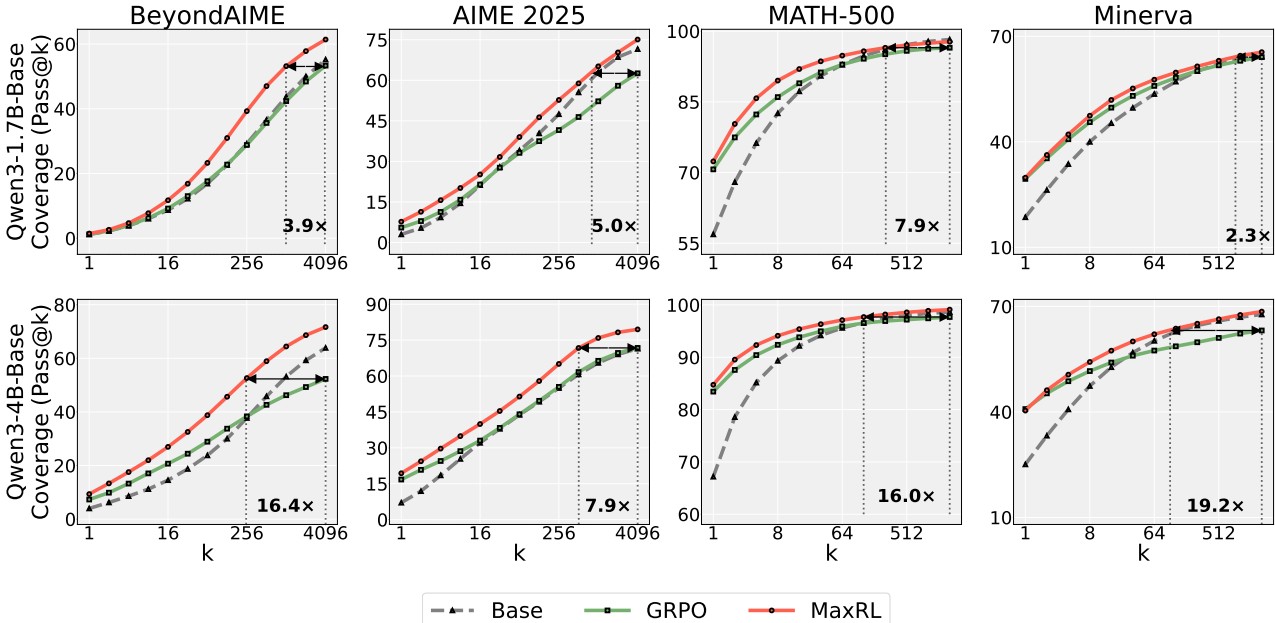

*Figure 5.* (**Qwen3 training results**) Evaluation of final checkpoints from training Qwen3-1.7B-Base and Qwen3-4B-Base models, on 4 benchmarks: AIME 2025, BeyondAIME, MATH-500 and Minerva. MAXRL matches or outperforms GRPO in all 4 evaluation datasets and shows little to no degradation at coverage (pass@k) for very high k values.

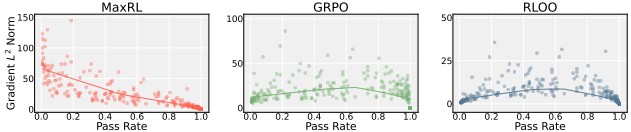

*Figure 6.* (**Gradient norm analysis**) To compare different objectives qualitatively, we show a scatter plot of gradient $L^2$ norm vs pass rate over individual prompts. We use Qwen2.5-1.5B-Instruct on MATH-500 dataset for this analysis. MAXRL generates larger gradient norms prompts with close to 0 pass rates.

16 rollouts per prompt, and 1000 RL steps. Notice that this setup utilizes lower rollout counts and RL steps than prior experiments to allow training larger models within our compute budget. We evaluate on four standard math benchmarks: AIME 2025, BeyondAIME (ByteDance-Seed, 2025), MATH-500 (Hendrycks et al., 2021; Lightman et al., 2024), and Minerva (Lewkowycz et al., 2022). We compare against GRPO (Shao et al., 2024), a widely used baseline for large-scale reasoning (Guo et al., 2025; Yang et al., 2025). Additional details are provided in Appendix K.

Figure 5 summarizes our main results. Across both model sizes, MAXRL consistently Pareto-dominates GRPO, achieving higher pass@1 while simultaneously improving pass@k. Consistent with prior work (Yue et al., 2025; Wu et al., 2026), GRPO exhibits pronounced pass@k degradation at larger $k$. In contrast, MAXRL improves pass@k relative to both the pretrained base model and the GRPO-trained checkpoint in 7 out of 8 evaluation settings. Improved pass@k directly translates into inference efficiency under repeated sampling. As shown in Figure 5, MAXRL

achieves up to $20\times$ test-time scaling efficiency gains when using a perfect verifier to filter wrong solutions, yielding substantial practical savings at inference time. We note that many settings admit a strong verifier, such as programming (Chen et al., 2021) or Lean (de Moura et al., 2015), where MAXRL can show strong benefits over other RL objectives. We provide additional results, such as evaluation on 4 additional benchmarks, and comparisons with additional baselines, in Appendix Sections K.3 and L.

> **Takeaway 4**
> On larger scale mathematical reasoning, MAXRL Pareto-dominates GRPO, shows little to no diversity degradation with respect to the base model, and leads to strong (up to $20\times$) test-time scaling efficiency gains.

### 6.5. Behavior Analysis of MAXRL

Finally, we study whether MAXRL shows different characteristics compared to commonly used RL objectives besides performance metrics. We first study the gradient norms produced by different objectives: Figure 6 illustrates that MAXRL generates higher gradient norms on harder prompts and lower gradient norms on easier prompts. This behavior matches that of cross-entropy in fully differentiable image classification setting (Figure 8), showing that MAXRL concentrates learning signal on harder problems unlike GRPO and RLOO. These larger gradient norms on more difficult prompts then translate into the model's capability to generate correct solutions for a larger fraction of problems during training. Figure 7 illustrates this trend, as MAXRL consistently generates at least one correct rollout for a larger frac-

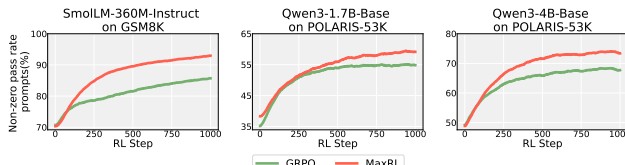

*Figure 7.* (**Training dynamics comparison**) Fraction of prompts where the model generates at least one correct rollout (out of 128, 16, and 16 rollouts for SmolLM-360M-Instruct, Qwen3-1.7B-Base, and Qwen3-4B-Base, respectively) during training. MAXRL consistently produces at least one correct rollout for more prompts.

tion of training prompts, and the gap between MAXRL and GRPO persists as we train longer. Appendix M.1 further corroborates these findings in the maze and GSM8K training settings. In addition, Figure 14 shows that GRPO and RLOO drive the per-prompt pass-rate distribution to the extremes of 0 and 1, consistent with the distribution-sharpening narrative, whereas MAXRL maintains a broad pass-rate distribution throughout training. Finally, the inverse signal-to-noise ratio (SNR) analysis in Figure 29 shows that MAXRL becomes more stable as the rollout budget increases, despite a stronger amplification of the learning signal from problems with very-low-success-rate ($p \to 0$). Appendix M further examines how MAXRL and GRPO differ in distribution sharpening and non-binary reward settings. In general, these results suggest that MAXRL differs in significant ways from other RL objectives, and we leave a deeper investigation of these differences to future work.

> **Takeaway 5**
>
> Besides performance metrics, MAXRL also exhibits different optimization dynamics. Most notably, it produces stronger gradients on harder prompts, and also leads to a larger fraction of prompts with at least one correct rollout during training.

## 7. Related Work

**Supervised training vs reinforcement learning.** Supervised learning and reinforcement learning (RL) are complementary but fundamentally different paradigms. Supervised training is stable, sample-efficient, and well-calibrated within the training distribution (Ng & Jordan, 2001), but it is limited by the quality and scope of available data and cannot directly optimize non-differentiable objectives such as correctness or preferences. In contrast, RL can optimize such objectives directly, typically via policy gradients (Williams, 1992; Sutton et al., 1999; 1998; Schulman et al., 2017; Guo et al., 2017), and improve performance beyond the available demonstrations by having access to interactions with an environment and the resulting reward-based feedback. Although recent work has reframed RL objectives as supervised ones (Rafailov et al., 2023), on-policy learning, characteristic of online RL algorithms, appears crucial for optimal performance (Tajwar et al., 2024; Xu et al., 2024). Modern foundation model training often combines supervised learning on human data with subsequent RL (Ouyang

et al., 2022). Unlike these approaches, we assume no access to high-quality demonstrations or a stronger model, and instead study a purely interactive RL setting that nonetheless optimizes an objective that mimics cross-entropy.

**Closely related work.** One closely related line of work is that of Xiong et al. (2025b), which also considers non-linear functions of the pass rate in reinforcement learning. Both their work and ours are motivated by the observation that expected-reward objectives can underweight low-pass-rate prompts, and that maximizing a log-likelihood-like objective can mitigate this issue. However, the focus of Xiong et al. (2025b) is on adaptive rollout budget allocation and sampling strategies, treating the choice of non-linear weighting as part of an algorithmic design space. In contrast, we do not employ adaptive sampling, and instead derive a sampling-based on-policy estimator that approaches maximum likelihood as rollout compute budget is increased. We also empirically focus on demonstrating better data and compute scaling with our framework, while Xiong et al. (2025b) focuses on comparing against other adaptive sampling frameworks (Yu et al., 2025). Furthermore, Davis & Recht (2025) provides a theoretical argument characterizing the population-level objectives approached by specific binary-reward reinforcement-learning algorithms in asymptotic limits, showing that certain procedures induce log-like weighting of the pass rate. Our work addresses a complementary question: how finite sampling defines explicit population-level objectives. We establish an exact estimator–objective equivalence at each finite rollout count and empirically evaluate how this objective-level interpolation manifests as compute is increased in both controlled settings and large-scale LLM post-training experiments. We discuss additional related work in Appendix B.

## 8. Conclusion

In this work, we establish Maximum Likelihood Reinforcement Learning as a principled optimization framework for non-differentiable binary reward settings. We showed that MAXRL approaches maximum likelihood in differentiable settings as compute increases and that in non-differentiable settings it offers key advantages over traditional RL, scaling more effectively with additional compute and data. More broadly, our results suggest that some limitations attributed to reinforcement learning with foundation models arise from objective choice rather than optimization or sampling. Our work currently assumes a binary reward setting and does not extend directly to continuous or arbitrarily valued rewards. Moreover, one can also consider objectives other than maximum likelihood to optimize following our framework. Generalizing MAXRL to continuous rewards, multi-turn reinforcement learning, and off-policy settings such as PPO-style training are promising directions for future work.

# Acknowledgements

This research used both the DeltaAI advanced computing and data resource (Bode et al., 2025), which is supported by the National Science Foundation (award OAC 2320345) and the State of Illinois, and the Delta advanced computing and data resource (Gropp et al., 2023), which is supported by the National Science Foundation (award OAC 2005572) and the State of Illinois. Delta and DeltaAI are joint efforts of the University of Illinois Urbana-Champaign and its National Center for Supercomputing Applications. Both of these computing resources were used as part of ACCESS-approved compute grants (Boerner et al., 2023). The authors also appreciate the computing resources of Bridges-2 (Brown et al., 2021) at Pittsburgh Supercomputing Center through ACCESS allocation CIS240901 from the Advanced Cyberinfrastructure Coordination Ecosystem: Services & Support (ACCESS) program, which is supported by National Science Foundation grants #2138259, #2138286, #2138307, #2137603, and #2138296. Overall, this project used ACCESS grants CIS240901, CIS250216, CIS250428, CIS250651, CIS250835, CIS250560, CIS251063, and CIS251385 for its compute resources. The authors also thank the CMU FLAME center and the CMU Babel Compute Cluster for additional compute support during the early phases of this project.

This work was supported in part by ONR N000142312368 and ONR MURI N00014-25-1-2116. Moreover, Fahim Tajwar was partially supported by the U.S. Army Futures Command under Contract No. W519TC-23-C-0030 during the project. Yuda Song gratefully acknowledges the support of the Two Sigma Fellowship. Daman Arora was supported by the National Science Foundation under Grants CCF-2106778. Yiding Jiang gratefully acknowledges the support of the Google PhD Fellowship.

The authors thank Brandon Pusateri, Jillian Lehosky, and Greg Bauer from ACCESS Support Staff for their incredible help in approving supplements and renewals for ACCESS compute grants throughout this project. Moreover, the work would not have been completed in a timely manner without the help of Brett Bode from NCSA Delta Support Staff, who provided the authors with critical help in properly utilizing the Delta cluster. The authors gratefully acknowledge Qianqian Wang, Yifei Zhou, Samuel Sokota, Yutong He, Lili Chen, Stephan Xie, Haque Ishfaq, Gaurav Rohit Ghosal, Aakash Lahoti, Kevin Li and other members of the Zanette, Russ and Auton lab for feedback and suggestions received on earlier versions of this work.

# Impact Statement

Our work can be used to train large language models that have better reasoning capabilities, which can have potential impact in the real world as LLM reasoning systems become widespread. Our experiments are conducted in relatively simple and controlled settings and it is an open question what kind of impacts large-scale LLM reasoning systems will have on society. Other than that, this paper presents work whose goal is to advance the field of Machine Learning. There are many potential overall societal consequences of our work, none of which we feel must be specifically highlighted here.

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

# Appendix

## A. Public Code Release

We provide sufficient details about our implementation, hyperparameters, environment design, and datasets in the main paper and the appendix to effectively reproduce the results in this paper. The assets required to reproduce the results in our paper can be found through our website: `https://zanette-labs.github.io/MaxRL/`. In particular, the code and scripts to reproduce the results in this paper has been open-sourced at `https://github.com/tajwarfahim/maxrl`.

## B. Extended Related Work

**Cross-entropy objective.** Maximizing log-likelihood via optimizing cross-entropy is a widely used framework in machine learning due to its simplicity and favorable theoretical properties. In particular, cross-entropy is a strictly proper scoring rule, meaning that the expected loss is uniquely minimized by the true probability distribution, which encourages statistically calibrated predictions (Good, 1952; Savage, 1971; Gneiting & Raftery, 2007; Waghmare & Ziegel, 2025). Moreover, it yields statistically efficient estimators under standard assumptions (van der Vaart, 1998; Casella & Berger, 2002; Lehmann & Casella, 2006), and induces gradients that concentrate learning signal on low-probability or uncertain outcomes through logarithmic weighting (Wang et al., 2020). As a result, cross-entropy (or log-loss) often yields consistent estimators for classification and tends to generalize well in practice (Ng & Jordan, 2001; Zhang, 2004). However, more recent work has shown that models trained to maximize log-likelihood can still overfit and exhibit miscalibration, motivating post-hoc techniques such as temperature scaling (Niculescu-Mizil & Caruana, 2005; Guo et al., 2017). Moreover, the unbounded nature of cross-entropy and its sensitivity to small perturbations in the predicted distribution suggest that alternative strictly proper scoring rules may be more suitable in certain settings (Kornblith et al., 2021). Because cross-entropy is extensively studied, we refer interested readers to Mao et al. (2023); Li et al. (2025); Terven et al. (2025) for a comprehensive review.

**Supervised training vs reinforcement learning.** Supervised learning has been the go-to training paradigm in machine learning, beginning with early "learning with a teacher" neural-network-based systems such as the perceptron (Rosenblatt, 1958) and later becoming practical with backpropagation-based neural network training (Rumelhart et al., 1986). It has been used to tackle a broad range of problems, from financial fraud detection (Afriyie et al., 2023; Editya et al., 2025), to sentiment analysis (Zhang et al., 2018c) and spam detection (Jain et al., 2022; Jamil et al., 2025). More recently, supervised training has been used in modern image classification systems (Lecun et al., 1998; Krizhevsky et al., 2012) to achieve strong performance. In modern foundation models, "pretraining" is typically done by supervised learning via minimizing cross-entropy over next-token prediction (Radford et al., 2018; 2019) on a large corpus of text, followed by additional supervised training on high-quality, human-written demonstrations (Ouyang et al., 2022; Zhang et al., 2023) to teach models how to respond to prompts, often known as instruction-tuning. In sequential decision-making domains, supervised training appears as behavior cloning from expert demonstration, used in early autonomous driving and robotic systems (Pomerleau, 1988; Bojarski et al., 2016; Codevilla et al., 2018). However, supervised learning in sequential decision-making breaks the i.i.d. assumption since the learned policy's actions affect which states it visits during execution, causing compounding errors over time (Ross & Bagnell, 2010; Belkhale et al., 2023). The classic no-regret reductions line (DAgger) makes this explicit and addresses drift by iteratively querying the expert on states visited by the learner, turning the sequential problem into an online supervised learning loop with improved guarantees (Ross et al., 2011).

In such scenarios, reinforcement learning is an attractive alternative paradigm that formalizes learning under delayed, sparse, and evaluative feedback in Markov decision processes (MDPs), with foundational roots in dynamic programming and MDP theory (Bellman, 1957). Mechanistically, the key contrast with supervised learning / behavior cloning is that supervised learning assumes (or benefits strongly from) an i.i.d. dataset of correct targets under a fixed data distribution, whereas RL's data distribution is policy-induced and nonstationary, and gradients arise from credit assignment through rewards rather than direct target labels. This mismatch shows up starkly in sequential prediction/imitation: naive behavior cloning trains on expert state distributions but at test time visits states induced by its own errors, causing compounding error (a form of distribution shift / "state drift"). Classical RL algorithms include temporal-difference learning (Sutton, 1988; Sutton et al., 1998) and value-based control such as Q-learning (Watkins & Dayan, 1992), while policy-gradient methods (Williams, 1992; Sutton et al., 1999) directly optimize expected return via likelihood-ratio gradients (REINFORCE) and later stabilized large-scale learning through variants like trust-region policy optimization (TRPO) (Schulman et al., 2015) and off-policy actor–critic methods for continuous control (e.g., DDPG (Lillicrap et al., 2016)) and maximum-entropy actor–critic (e.g., SAC (Haarnoja et al., 2018)). Empirically, deep RL's modern resurgence is often associated with representation learning + RL on high-dimensional inputs (e.g., DQN (Mnih et al., 2013)).

A large body of work blends supervised and RL to get the best of both: (i) Imitation + online correction methods like DAgger (Ross et al., 2011) explicitly combine supervised learning with interactive data collection to mitigate distribution shift ; (ii) Inverse RL / MaxEnt IRL reframes imitation as learning a reward/cost model that explains expert behavior, with maximum-entropy formulations giving a principled probabilistic objective (Ziebart et al., 2008); (iii) Adversarial imitation (GAIL) (Ho & Ermon, 2016) avoids explicit reward learning by matching occupancy measures via a GAN-like discriminator, typically trained with policy optimization; (iv) Learning from demonstrations in deep RL injects supervised losses and/or demonstration replay into RL to improve exploration and sample efficiency—e.g., DQfD (Hester et al., 2018) combines TD learning with supervised large-margin imitation terms, and demonstration-augmented continuous-control methods address sparse-reward exploration failures (Nair et al., 2018); and (v) Trajectory-optimization–guided policy learning (guided policy search (Levine & Koltun, 2013)) explicitly produces supervised targets for a policy network from trajectory optimization / local controllers, bridging optimal control, RL, and supervised regression. In modern LLM alignment, the same hybrid template appears as "Supervised fine-tuning + preference-based RL": InstructGPT (Ouyang et al., 2022) first performs supervised fine-tuning on demonstrations and then applies RL from human feedback (RLHF), while more recent approaches like Direct Preference Optimization (DPO) (Rafailov et al., 2023) recast parts of RLHF into a supervised-style classification objective — illustrating an active trend of recovering supervised-like training signals even when the underlying goal is preference/reward optimization. Concurrent work such as (Russo, 2026) has also studied policy improvement through the lens of success conditioning. Another closely related view for comparing REINFORCE and Maximum Likelihood is through the KL divergence: where REINFORCE can be seen as a reverse KL and Maximum Likelihood can be seen as forward KL objective with different properties, as noted by Chan et al. (2022). Thus MAXRL is also conceptually related to RWR (Peters & Schaal, 2007)/AWR (Peng et al., 2019), which are also forward KL objectives.

Control as inference is also a closely related topic (Millidge et al., 2020; O'Donoghue et al., 2020; Rawlik et al., 2012; Ito & Kashima, 2024; Tarbouriech et al., 2023), and we point the reader to Levine (2018) for more details. Finally, our approach is closely related to Monte Carlo Expectation Maximization (MCEM), which approximates intractable expectations in maximum-likelihood estimation using samples (Dempster et al., 1977; Wei & Tanner, 1990; Levine & Casella, 2001; Ruth, 2024). In the infinite sample regime, MAXRL can be viewed through an EM lens: the observed event is that the answer is correct, and the latent variable is the successful trajectory that produced it. A sample-until-success implementation of this maximum-likelihood objective would therefore recover an MCEM-style procedure. The finite-compute version of MAXRL, however, makes a different design choice: it fixes the number of rollouts and thereby defines a compute-indexed family of sample-based objectives, interpolating between standard reward maximization and exact maximum likelihood. Thus, rather than using Monte Carlo samples only to approximate a fixed EM objective, MAXRL uses the sampling budget itself to define the learning objective optimized by policy-gradient methods.

**Training LLMs for strong reasoning abilities.**    A few different approaches for post-training have demonstrated success, including supervised fine-tuning on human-crafted, high-quality demonstrations (Wang et al., 2023b), iterative supervised training on self-generated, good-quality responses (Anthony et al., 2017; Zelikman et al., 2022; Gulcehre et al., 2023), reinforcement learning from a learned reward model on human preferences (Ouyang et al., 2022), and more recently preference-based contrastive learning (Rafailov et al., 2023; Pang et al., 2024). Reinforcement learning from verifiable rewards (RLVR), where LLMs receive reward from a ground truth verifier instead of using a trained reward model, has emerged as the dominant paradigm for instilling strong reasoning capabilities into LLMs (OpenAI et al., 2024; Guo et al., 2025; Team et al., 2025; Lambert et al., 2025a; Yang et al., 2025). Whereas supervised training learns better behavior from fixed static datasets, reinforcement learning uses policy gradient algorithms (e.g., PPO (Schulman et al., 2017), GRPO (Shao et al., 2024), RLOO (Ahmadian et al., 2024)) to learn from self-generated responses and non-differentiable rewards. However, these algorithms and their variants (Zheng et al., 2025; Liu et al., 2025b; MiniMax et al., 2025) optimize expected reward or pass rate and only differ in how the advantage or off-policy updates are calculated. In contrast, in our work we focus on recovering the cross-entropy-based classification objective in an RL training pipeline, fundamentally differing from previous work.

Since the recent advent of RLVR, the RLVR pipeline has been studied extensively (Zeng et al., 2025b; Liu et al., 2025b; Khatri et al., 2026) and alternative algorithms such as Dr.GRPO (Liu et al., 2025b), DAPO (Yu et al., 2025), GSPO (Zheng et al., 2025), CISPO (MiniMax et al., 2025), and RAFT (Xiong et al., 2025a) have been proposed. The idea of normalizing advantages by mean reward, similar to ours, has been explored in Huang et al. (2025), but whereas we normalize advantage by group mean reward (mean reward over the rollouts associated with a particular prompt), Huang et al. (2025) normalizes by the batch mean reward (mean reward over all prompts in a batch of policy gradient updates). A limitation of our work is that we assign no gradient for tasks where the policy fails to generate even one correct rollout: recent work such as Feng

et al. (2025) has studied this problem by employing a maximum-likelihood (MLE) objective in reward modeling to assign non-zero, confidence-dependent rewards to incorrect generations. Similarly to us, earlier work has studied how to learn effectively via RL from a collection of tasks as opposed to a single task, where each task may have different difficulties, and thus reward distribution. PopArt (Hessel et al., 2019) is one such example, which uses a normalization by standard deviation for the rewards of each task (as opposed to normalization by mean reward in MAXRL), and we consider this as an earlier example of normalization by standard deviation before GRPO (Shao et al., 2024). Finally, recent work such as Zhang et al. (2026) has also studied how RL training is influenced by pretraining and midtraining in toy didactic settings, establishing the importance of good pretraining/midtraining for the success of RL, similar to Gandhi et al. (2025).

**RL training causes distribution sharpening.** Despite its usefulness, questions remain on whether RLVR teaches LLMs fundamentally new behavior/skills, or simply sharpens existing good behavior from the pretrained model. Previous research (Liu et al., 2025b; Zhao et al., 2025; AI et al., 2025) demonstrated that certain reasoning skills like reflection already exist in the pretrained model, and Gandhi et al. (2025) shows that good reasoning behaviors learned from pretraining are crucial for the success of RLVR in the post-training phase. More recently, studies (Yue et al., 2025; Dang et al., 2025; Wu et al., 2026) found that RLVR decreases the model's diversity by reducing pass@k. In our paper, we confirm these findings and attribute this to the RL objective itself as optimizing expected reward tends to marginalize learning signal from harder prompts, which results in distribution sharpening.

**On exploration for reinforcement learning for LLMs.** Exploration, or taking actions to discover new information, is a widely studied topic in reinforcement learning. A closely related topic is *curiosity*, where an agent seeks new information about its environment via interactions. *Intrinsic motivation* is a popular notion for curiosity, where the agent is driven by an exploration bonus that is not necessarily related to the task to be achieved (Schmidhuber, 1991; 2007). Follow-up research has built on this notion to mitigate problems of sparse reward (reward is observed at a very belated phase of interactions) or no reward at all (Pathak et al., 2017; 2019; Eysenbach et al., 2018; Burda et al., 2018; Sharma et al., 2019; Yang et al., 2024c; Houthooft et al., 2016). Count-based bonuses have also been introduced as a way of computing intrinsic motivation (Bellemare et al., 2016). Prompt-level reweighting of gradients has also been studied (Yu et al., 2026), though under a different context (self-training) and a different weighting mechanism. Finally, adding noise to network parameters or optimization has been another line of work to improve exploration during RL training (Fortunato et al., 2018; Ishfaq et al., 2024b;a; 2025). Maximum entropy RL, the principle where one attempts to recover an agent that achieves high reward but is as stochastic as possible, can be seen as another attempt at solving exploration for classical RL (Haarnoja et al., 2018; Boucher et al., 2025; Eysenbach & Levine, 2022; Dong et al., 2025). In summary, exploration-exploitation tradeoff (Sutton, 1988; Auer et al., 2002; Thompson, 1933) has been a crucial topic for ensuring RL agents' success.

More recently, exploration has emerged as an important topic for building modern LLM-based systems. There are two types of exploration to consider. The first is *inference-time exploration*, where an agent has to efficiently gather information during deployment by strategically choosing its interactions with its environment, Tajwar et al. (2025) is an important work in this line of research. More importantly, pass@k degradation (mode collapse) during RLVR (Yue et al., 2025; Wu et al., 2026; GX-Chen et al., 2026) has prompted research into *train-time exploration*, where the challenge is to go beyond the pretrained model's capabilities (Ni et al., 2025) and discover new knowledge. Primary approaches include directly optimizing for pass@k (Mahdavi et al., 2025; Walder & Karkhanis, 2025; Tang et al., 2025; Chen et al., 2025b), curriculum learning (Tajwar et al., 2025; Chen et al., 2025a; Setlur et al., 2026; Motwani et al., 2026), learning from additional hints or abstractions (Qu et al., 2026; Chen et al., 2026a; Liu et al., 2026), increasing number of rollouts to prevent RL gains from saturating (Hu et al., 2026), employing data curation algorithm to redirect effort to problems with low success rate (Nguyen et al., 2025b), leveraging expert guidance (Chang et al., 2024; Qu et al., 2025), or differential smoothing by penalizing entropy on low reward trajectories and encouraging entropy on high reward trajectories. Entropy-based bonuses to encourage exploration during RL training (Hao et al., 2025; Chen et al., 2026b; Cheng et al., 2026; Wang et al., 2025; Petrenko et al., 2026; Ged & Veiga, 2024) is another popular line of work for improving exploration. A few modern approaches for exploration bonus utilized for LLM training are proposed by Song et al. (2025); Tuyls et al. (2026). The idea of curiosity-driven exploration from classical RL discussed above has also been adopted for LLMs (Dai et al., 2026). Although some literature has reported pass@k degradation during RL training, others have found the opposite results. For example, ProRL (Liu et al., 2025a) has shown that RL training on a mixture of reasoning puzzles (Stojanovski et al., 2025) can improve pass@k on a heldout reasoning task. Similarly, Yuan et al. (2026) has shown that LLMs can learn new skills via RL by composing old ones, showing the promise of going beyond pretraining knowledge, and Cheng et al. (2025) also found pass@k to improve, particularly on tasks less likely to appear during the pretraining stage. Recent work also formalized Set-RL (Hamid et al.,

2026; Orney et al., 2026) and Risk-Sensitive Reinforcement Learning (RS-GRPO) (Jiang et al., 2026) to improve diversity during LLM reasoning training and better navigate the exploration-exploitation tradeoff in LLM RL training, with the latter being a closely related prior work for our own work. A parallel effort has gone into difficulty-aware adaptive sampling, with Xiong et al. (2025b); Nguyen et al. (2025b); Yang et al. (2026b) being notable examples. Ray interference (Schaul et al., 2019) has been proposed as an explanation for the observed pass@k degradation. Overall, this line of research remains important as focus moves to LLMs discovering new information during RL training and it is therefore an ongoing field of research.

**Generalization in reinforcement learning.** Research on generalization in reinforcement learning asks a core question: does an agent learn principles that transfer beyond the exact environments it trained in, or does it just memorize experience (Zhang et al., 2018b;a; Schaul et al., 2019; Bengio et al., 2020)? Empirical work shows deep RL agents often overfit to training seeds, visuals, or dynamics, performing poorly on new levels, layouts, or slightly shifted physics. To study this, researchers built procedural benchmarks (like CoinRun/Procgen (Cobbe et al., 2020)) and multi-task suites (e.g., robotics task collections (Yu et al., 2019; Atamuradov, 2025) or LLM sequential decision-making task suites (Tajwar et al., 2025)) that separate train and test environments. A major line of work improves generalization through regularization (Kostrikov et al., 2021) and invariances (Zhang et al., 2021) — especially data augmentation (Laskin et al., 2020; Raileanu et al., 2021), mixup-style methods (WANG et al., 2020), and representation learning tricks (Higgins et al., 2017; Srinivas et al., 2020; Wang et al., 2021) that make policies rely less on superficial visual details. Another branch focuses on task and domain shift, using meta-RL (Duan et al., 2016; Finn et al., 2017; Liang et al., 2024), multi-task learning (Brunskill & Li, 2013), domain randomization, and distributionally robust RL (Clavier et al., 2022; Lu et al., 2024; Shi et al., 2023) to handle new tasks or uncertain dynamics. Recent lines of work have also directly studied exploration as a means to achieve generalization in RL (Jiang et al., 2023), and have shown that simple architectural changes and scale can often improve generalization in the ProcGen benchmark (Jesson & Jiang, 2024). On the theory side, classical PAC-MDP (Strehl et al., 2009) and robust MDP frameworks (Nilim & Ghaoui, 2004; Iyengar, 2005) formalize when policies learned from limited samples can be expected to work in new situations. In the LLM settings, analogous questions appear in RLHF (Ouyang et al., 2022), where reinforcement learning is used to align models with human preferences, and researchers now study how this training affects generalization to unseen prompts, behaviors, and user distributions (Kirk et al., 2024; Lin et al., 2024; Lambert et al., 2025b; Jia, 2024; Li et al., 2026). Overall, the field has moved from "can RL learn?" to "what exactly does it learn, and when does that knowledge transfer?"

# C. Theoretical Results

Here we present the proofs of theorems mentioned in the main paper. First we restate and prove Theorem 4.1.

**Theorem C.1** (Restatement of Theorem 4.1). *The gradient of the maximum likelihood objective admits the following conditional expectation representation:*

$$\nabla_\theta J_{\mathrm{ML}}(x) = \mathbb{E}[\nabla_\theta \log m_\theta(z \mid x) \mid f(z) = y^*(x)].$$

*Proof.* Recall the standard REINFORCE identity for the gradient of the pass rate:

$$\nabla_\theta p_\theta^{\mathrm{pass}}(x) = \nabla_\theta \mathbb{E}_{z \sim m_\theta(\cdot \mid x)}[\mathbb{I}\{f(z) = y^*(x)\}] = \mathbb{E}_{z \sim m_\theta(\cdot \mid x)}[\mathbb{I}\{f(z) = y^*(x)\}\nabla_\theta \log m_\theta(z \mid x)].$$

The gradient of the maximum likelihood objective is:

$$\nabla_\theta J_{\mathrm{ML}}(x) = \nabla_\theta \log p_\theta^{\mathrm{pass}}(x) = \frac{\nabla_\theta p_\theta^{\mathrm{pass}}(x)}{p_\theta^{\mathrm{pass}}(x)} = \frac{\mathbb{E}_{z \sim m_\theta(\cdot \mid x)}[\mathbb{I}\{f(z) = y^*(x)\}\nabla_\theta \log m_\theta(z \mid x)]}{\mathbb{E}_{z \sim m_\theta(\cdot \mid x)}[\mathbb{I}\{f(z) = y^*(x)\}]}.$$

By the definition of conditional expectation for an event $A$ with $\mathbb{P}(A) > 0$:

$$\mathbb{E}[X \mid A] = \frac{\mathbb{E}[X \cdot \mathbb{I}_A]}{\mathbb{P}(A)}.$$

Letting $X = \nabla_\theta \log m_\theta(z \mid x)$ and $A = \{z : f(z) = y^*(x)\}$, and noting that $p_\theta^{\mathrm{pass}}(x) = \mathbb{P}(A)$, we obtain:

$$\nabla_\theta J_{\mathrm{ML}}(x) = \mathbb{E}[\nabla_\theta \log m_\theta(z \mid x) \mid f(z) = y^*(x)].$$

$\square$

Next, we restate and prove Theorem 4.2.

**Theorem C.2** (Restatement of Theorem 4.2). *The estimator $\widehat{g}_N(x)$ is an unbiased estimator for the* MAXRL *gradient of order $T = N$, i.e.,*

$$\mathbb{E}[\widehat{g}_N(x)] = \nabla_\theta J_{\mathrm{MAXRL}}^{(N)}(x).$$

*Proof.* Conditioned on $K \geq 1$, the successful samples are i.i.d. draws from the success-conditioned distribution, so by Theorem 4.1:

$$\mathbb{E}[\widehat{g}_N(x) \mid K \geq 1] = \nabla_\theta \log p_\theta^{\mathrm{pass}}(x).$$

Since $\widehat{g}_N(x) = 0$ when $K = 0$:

$$\mathbb{E}[\widehat{g}_N(x)] = \nabla_\theta \log p_\theta^{\mathrm{pass}}(x) \cdot \mathbb{P}(K \geq 1) = \nabla_\theta \log p_\theta^{\mathrm{pass}}(x) \cdot \mathrm{pass@}N(x).$$

Writing $p = p_\theta^{\mathrm{pass}}(x)$ and using $\mathrm{pass@}k(x) = 1 - (1-p)^k$:

$$\frac{\nabla_\theta p}{p} \cdot (1 - (1-p)^N) = \nabla_\theta p \sum_{k=1}^{N}(1-p)^{k-1} = \sum_{k=1}^{N} \frac{1}{k}\nabla_\theta \mathrm{pass@}k(x) = \nabla_\theta J_{\mathrm{MAXRL}}^{(N)}(x),$$

where the second equality uses $\nabla_\theta \mathrm{pass@}k(x) = k(1-p)^{k-1}\nabla_\theta p$.

$\square$

## D. Analysis of the Practical Gradient Estimator from Algorithm 1

In this section, we will analyse the practical gradient estimator from Algorithm 1, which includes both an unconditional average score baseline to reduce variance, and sets the gradient to be 0 in the case of all negative rollouts (no baseline is used). We show that this changes the objective for which the gradient is unbiased for: while the gradient from Equation (9) is unbiased for the MAXRL gradient of order $T = N$ (Theorem 4.2), the practical gradient estimator is instead unbiased for the MAXRL gradient of order $T = N - 1$. We formally prove this result below.

**Theorem D.1** (Effective truncation of the dropped-baseline estimator). *Fix an input $x$ and let $z_1, \ldots, z_N \sim m_\theta(\cdot \mid x)$ be $N$ i.i.d. rollouts from the latent policy. Define*

$$r_i = \mathbb{I}\{f(z_i) = y^*(x)\}, \qquad K = \sum_{i=1}^{N} r_i, \qquad S_i = \nabla_\theta \log m_\theta(z_i \mid x).$$

*Let*

$$p_\theta^{\text{pass}}(x) = \mathbb{P}_{z \sim m_\theta(\cdot \mid x)}(f(z) = y^*(x)),$$

*and assume the standard score-function identities*

$$\mathbb{E}[S_i] = 0, \qquad \mathbb{E}[r_i S_i] = \nabla_\theta p_\theta^{\text{pass}}(x).$$

*Consider the practical gradient estimator from Algorithm 1*

$$\widetilde{g}_N(x) = \mathbb{I}\{K > 0\} \left( \frac{1}{K} \sum_{i=1}^{N} r_i S_i - \frac{1}{N} \sum_{i=1}^{N} S_i \right).$$

*Then*

$$\mathbb{E}[\widetilde{g}_N(x)] = \sum_{t=1}^{N-1} (1 - p_\theta^{\text{pass}}(x))^{t-1} \nabla_\theta p_\theta^{\text{pass}}(x).$$

*Equivalently, the practical dropped-baseline estimator is unbiased for the $(N-1)$-truncated MAXRL gradient.*

*Proof.* For readability, write $p = p_\theta^{\text{pass}}(x)$ and $\nabla p = \nabla_\theta p_\theta^{\text{pass}}(x)$. Decompose the estimator as

$$\widetilde{g}_N(x) = \underbrace{\mathbb{I}\{K > 0\} \frac{1}{K} \sum_{i=1}^{N} r_i S_i}_{\widetilde{g}_N^{\text{succ}}(x)} - \underbrace{\mathbb{I}\{K > 0\} \frac{1}{N} \sum_{i=1}^{N} S_i}_{\widetilde{g}_N^{\text{base}}(x)}.$$

By the standard MAXRL estimator identity from Theorem 4.2, the successful-rollout term satisfies

$$\mathbb{E}[\widetilde{g}_N^{\text{succ}}(x)] = \sum_{t=1}^{N} (1 - p)^{t-1} \nabla p.$$

It remains to compute the expectation of the dropped baseline term. Since

$$\mathbb{E}\left[ \frac{1}{N} \sum_{i=1}^{N} S_i \right] = 0,$$

we have

$$\mathbb{E}\left[ \mathbb{I}\{K > 0\} \frac{1}{N} \sum_{i=1}^{N} S_i \right] = -\mathbb{E}\left[ \mathbb{I}\{K = 0\} \frac{1}{N} \sum_{i=1}^{N} S_i \right].$$

Now observe that

$$\mathbb{I}\{K = 0\} = \prod_{\ell=1}^{N} (1 - r_\ell).$$

Using independence of the samples,

$$
\mathbb{E}\left[\mathbb{I}\{K=0\}\frac{1}{N}\sum_{i=1}^{N}S_i\right] = \frac{1}{N}\sum_{i=1}^{N}\mathbb{E}[(1-r_i)S_i]\prod_{\ell\neq i}\mathbb{E}[1-r_\ell]
$$

$$
= \frac{1}{N}\sum_{i=1}^{N}\left(\mathbb{E}[S_i]-\mathbb{E}[r_iS_i]\right)(1-p)^{N-1}
$$

$$
= -(1-p)^{N-1}\nabla p.
$$

Therefore,

$$
\mathbb{E}\left[\widetilde{g}_N^{\mathrm{base}}(x)\right] = (1-p)^{N-1}\nabla p.
$$

Combining the two terms gives

$$
\mathbb{E}[\widetilde{g}_N(x)] = \left[\sum_{t=1}^{N}(1-p)^{t-1}\nabla p\right] - (1-p)^{N-1}\nabla p
$$

$$
= \sum_{t=1}^{N-1}(1-p)^{t-1}\nabla p.
$$

Substituting back $p = p_\theta^{\mathrm{pass}}(x)$ yields

$$
\mathbb{E}[\widetilde{g}_N(x)] = \sum_{t=1}^{N-1}(1-p_\theta^{\mathrm{pass}}(x))^{t-1}\nabla_\theta p_\theta^{\mathrm{pass}}(x),
$$

which is exactly the $(N-1)$-truncated MAXRL gradient, concluding our proof. $\square$

# E. More on Unifying Weight-Function View on RL Objectives

Here we provide full derivations following Section 5. Recall that we want to express population-level gradients of different objectives in the following form:

$$\nabla_\theta J = \mathbb{E}_{x \sim \rho}\left[w(p_\theta(x))\nabla_\theta p_\theta(x)\right]$$

where $p_\theta(x) = p_\theta^{\text{pass}}(x)$ and $w(p_\theta(x))$ is the weighting function. In this section, we show that all objectives of our consideration can be written in this form. Furthermore, we will derive the weighting function $w(p_\theta(x))$ for each of them.

**Classical RL (REINFORCE).** For classical reinforcement learning, i.e., the REINFORCE objective, we have:

$$\begin{aligned}
J_{\text{RL}} &= \mathbb{E}_{x \sim \rho}[\mathbb{E}_{z \sim m_\theta(\cdot|x)}[r(x, z)]] \\
&= \mathbb{E}_{x \sim \rho}[\mathbb{E}_{z \sim m_\theta(\cdot|x)}[\mathbb{I}\{f(z) = y^*(x)\}]] \\
&= \mathbb{E}_{x \sim \rho}[p_\theta^{\text{pass}}(x)]
\end{aligned}$$

Therefore, its gradient is:

$$\nabla_\theta J_{\text{RL}} = \mathbb{E}_{x \sim \rho}[\nabla_\theta p_\theta^{\text{pass}}(x)]$$

giving the corresponding $w_{\text{RL}}$ to be 1.

**GRPO.** Our analysis is similar to that of Davis & Recht (2025). The gradient of the population level GRPO objective gradient can be written as:

$$\nabla_\theta J_{\text{GRPO}} = \mathbb{E}_{x \sim \rho}\left[\mathbb{E}_{z \sim m_\theta(\cdot|x)}\left[\left(\frac{r(x, z) - \mathbb{E}_{z \sim m_\theta(\cdot|x)}[r(x, z)]}{\sqrt{\text{Var}_{z \sim m_\theta(\cdot|x)}[r(x, z)]}}\right) \nabla_\theta \log m_\theta(z|x)\right]\right]$$

Since we consider a binary reward setting, we have $\mathbb{E}_{z \sim m_\theta(\cdot|x)}[r(x, z)] = p_\theta^{\text{pass}}(x)$. Similarly, considering the variance of a Bernoulli random variable, we get:

$$\text{Var}_{z \sim m_\theta(\cdot|x)}[r(x, z)] = p_\theta^{\text{pass}}(x)\left(1 - p_\theta^{\text{pass}}(x)\right)$$

Therefore, the objective becomes:

$$\nabla_\theta J_{\text{GRPO}} = \mathbb{E}_{x \sim \rho}\left[\frac{1}{\sqrt{p_\theta^{\text{pass}}(x)\left(1 - p_\theta^{\text{pass}}(x)\right)}}\nabla_\theta p_\theta^{\text{pass}}(x)\right]$$

which thereby gives us the weighting function to be $1/\sqrt{p_\theta(x)(1 - p_\theta(x))}$, as desired.

**Maximum Likelihood (ML).** The maximum likelihood objective is given by

$$J_{\text{ML}} = \mathbb{E}_{x \sim \rho}[\log p_\theta^{\text{pass}}(x)].$$

Taking its gradient with respect to $\theta$ and applying the chain rule, we obtain

$$\begin{aligned}
\nabla_\theta J_{\text{ML}} &= \mathbb{E}_{x \sim \rho}\left[\nabla_\theta \log p_\theta^{\text{pass}}(x)\right] \\
&= \mathbb{E}_{x \sim \rho}\left[\frac{1}{p_\theta^{\text{pass}}(x)}\nabla_\theta p_\theta^{\text{pass}}(x)\right]
\end{aligned}$$

This shows that the weighting function for the maximum likelihood objective is $1/p_\theta^{\text{pass}}(x)$, as we claimed in Table 2.

**MAXRL.** Finally, we consider the objective $J_{\text{MAXRL}}^{(T)}$.

**Proposition E.1.** *For MAXRL with order $T$, we can rewrite it as*

$$\nabla_\theta J_{\text{MAXRL}}^{(T)} = \mathbb{E}_{x \sim \rho}[w(p_\theta(x))\ \nabla_\theta p_\theta(x)],$$

*where*

$$w_T(p) = \sum_{k=1}^{T}(1 - p)^{k-1} = \frac{1 - (1 - p)^T}{p}.$$

*Proof.* From Equation (7), we have:

$$\nabla_\theta J_{\text{MAXRL}}^{(T)}(x) = \sum_{k=1}^{T} \frac{1}{k} \nabla_\theta \text{pass@}k(x).$$

Using $\text{pass@}k(x) = 1 - (1-p)^k$ where $p = p_\theta^{\text{pass}}(x)$:

$$\nabla_\theta \text{pass@}k(x) = k(1-p)^{k-1} \nabla_\theta p.$$

Substituting:

$$\nabla_\theta J_{\text{MAXRL}}^{(T)}(x) = \sum_{k=1}^{T} \frac{1}{k} \cdot k(1-p)^{k-1} \nabla_\theta p = \left( \sum_{k=1}^{T} (1-p)^{k-1} \right) \nabla_\theta p = w_T(p) \nabla_\theta p_\theta^{\text{pass}}(x).$$

Taking the expectation over $x \sim \rho$ completes the proof. $\square$

# F. RL Implementation Details for LLM Experiments

Our discussion here follows that of Shafayat et al. (2025). For continuity with existing literature, we use slightly different notations from the rest of the paper for this section. Let $x$ represent a prompt, and let $y \sim \pi(\cdot|x)$ represent sequence of tokens autoregressively sampled from the language model $\pi$ conditioned on the prompt $x$. Let $\pi_\theta$ be the current policy, and $\pi_{\theta_{\text{old}}}$ be an older policy (from earlier iterations in training) used for data generation. In our implementation (based on verl (Zhang et al., 2024; Sheng et al., 2024)), we use the following general RL objective:

$$\mathcal{J}(\theta) = \mathbb{E}_{x \sim \mathcal{D}, \{y_i\}_{i=1}^G \sim \pi_{\theta_{\text{old}}}(\cdot|x)} \left[ \frac{1}{T} \sum_{i=1}^G \sum_{t=1}^{|y_i|} \min \left( w_{i,t}(\theta)\hat{A}_{i,t}, \text{clip}(w_{i,t}(\theta),\, 1-\varepsilon,\, 1+\varepsilon)\hat{A}_{i,t} \right) \right]$$

where $T$ is the total number of tokens in the mini-batch (excluding tokens in the prompt etc., since we only compute loss on the model generated tokens), $\pi_\theta$ represents the current LLM's autoregressive probability distribution, $\pi_{\theta_{\text{old}}}$ denotes the behavior policy/data generation policy's probability distribution, $w_{i,t}(\theta)$ is the importance ratio, defined as:

$$w_{i,t}(\theta) = \frac{\pi_\theta(y_{i,t} \mid x,\ y_{i,<t})}{\pi_{\theta_{\text{old}}}(y_{i,t} \mid x,\ y_{i,<t})}$$

Since we operate fully on-policy, i.e., one RL step per one batch of generated rollouts, this is always one in our experiments, and the clipping parameter $\epsilon$ has no effect on our training. $\hat{A}_{i,t}$ represents the advantage for the $t$-th token in the sequence $y_i$. The same advantage defined at a sequence level is applied to each token in the sequence, so henceforth we will drop the $t$ from the notation as well.

The main difference between GRPO (Shao et al., 2024), RLOO (Ahmadian et al., 2024) and MAXRL comes from their use of different advantage functions. RLOO objective uses the following advantage function:

$$\frac{1}{G} \sum_{i=1}^G [R(y_{(i)}, x) - \frac{1}{G-1} \sum_{j \neq i} R(y_{(j)}, x)]$$

whereas GRPO uses the following advantage function:

$$\hat{A}_i = \frac{r(x, y_i) - \text{mean}\left(\{r(x, y_i)\}_{i=1}^G\right)}{\text{std}\left(\{r(x, y_i)\}_{i=1}^G\right) + \epsilon}$$

where $\epsilon$ is a small number ($1 \times 10^{-6}$) added to avoid division by zero. Finally, the advantage for MAXRL is as follows:

$$\hat{A}_i = \frac{r(x, y_i) - \text{mean}\left(\{r(x, y_i)\}_{i=1}^G\right)}{\text{mean}\left(\{r(x, y_i)\}_{i=1}^G\right) + \epsilon}$$

Here $G$ is the number of online samples generated. RLOO, GRPO and MAXRL create a dynamic baseline for each sample without needing a separate value function (unlike PPO (Schulman et al., 2017)), effectively estimating the expected return on-the-fly during training. Not having a value network makes the training much simpler for all three algorithms.

# G. Pass@k Calculation

## G.1. Closed-form Calculation for Differentiable Settings

To calculate pass@k from a generative model, one usually samples $T \geq k$ rollouts from the model, calculate success or failure from each of them, and then uses an appropriate statistical estimator for pass@k (Chen et al., 2021; Yue et al., 2025). However, since there is no latent reasoning process involved in our didactic ImageNet experiments and since we can directly calculate the model likelihood of label $y \in \mathcal{Y}$ for an input image $x \in \mathcal{X}$, namely $\pi_\theta(y|x)$, we can also analytically compute pass@k without sampling as well. Formally, in all ImageNet experiments, we calculate pass@k for an example (image, label) pair $(x, y^*(x))$ as follows:

$$\text{Pass@}k(x, y^*(x); \pi_\theta) = 1 - (1 - \pi_\theta(y^*(x)|x))^k$$

The average pass@k is then obtained by averaging the above quantity over all example pairs in the validation dataset.

## G.2. Sampling-based Estimator

Unlike the ImageNet setting, we can't usually directly calculate pass@k via accessing the true probability of the correct action. Therefore, we use the default pass@k calculation mechanism in `verl` (Sheng et al., 2024; Zhang et al., 2024), using the bootstrapping low variance unbiased estimator introduced by Chen et al. (2021). This employs generating $n \geq k$ samples per task, counting the number of correct samples $c(x)$ among the $n$ samples, and estimate pass@k as:

$$\text{Pass@}k = \mathbb{E}_{x \sim \rho} \left[ 1 - \frac{\binom{n - c(x)}{k}}{\binom{n}{k}} \right]$$

# H. ImageNet Experiments

## H.1. Training Procedure

Let $\mathcal{X}$ be the input space and $\mathcal{Y}$ be the label space. Let $\pi_\theta$ denote our model: given an input image $x \in \mathcal{X}$, $\pi_\theta(y|x)$ is the model's predicted probability of image $x$ belonging to class $y \in \mathcal{Y}$. For an input image and label pair $(x, y^*(x))$, the cross-entropy loss is:

$$\mathcal{L}_{\text{CE}}(x, y^*(x); \pi_\theta) = -\log \pi_\theta(y^*(x)|x)$$

On the other hand, the corresponding RL objective for the same pair is:

$$\mathcal{L}_{\text{RL}}(x, y^*; \pi_\theta) = -\mathbb{E}_{y \sim \pi_\theta(\cdot|x)}[-\log \pi_\theta(y|x) \cdot \hat{A}(y|x)]$$

where the expectation is computed using Monte-Carlo sampling with $K$ rollouts of $y$ from $\pi_\theta(\cdot|x)$. GRPO, REINFORCE and MAXRL vary only in the calculation of the advantage $A(y|x)$. Concretely, let $y^{(1)}, \ldots, y^{(K)}$ be our $K$ rollouts, sampled from the conditional probability distribution $\pi_\theta(\cdot|x)$. We operate under a binary reward setting, meaning the reward function $r(x, y)$ is:

$$r(x, y) = \mathbb{I}[y = y^*(x)] = \begin{cases} 1, & \text{if } y = y^*(x) \\ 0, & \text{otherwise} \end{cases}$$

Given this reward, we calculate advantage under GRPO, REINFORCE and MAXRL as follows:

$$\hat{A}_{\text{GRPO}}(x, y) = \frac{r(x, y) - \hat{\mu}}{\hat{\sigma}}$$

$$\hat{A}_{\text{REINFORCE}}(x, y) = r(x, y) - \hat{\mu}$$

$$\hat{A}_{\text{MAXRL}}(x, y) = \frac{r(x, y) - \hat{\mu}}{\hat{\mu}}$$

where $\hat{\mu} = \frac{\sum_{i=1}^{K} r(x, y^{(i)})}{K}$, $\hat{\sigma} = \sqrt{\frac{\sum_{i=1}^{K} (r(x, y^{(i)}) - \hat{\mu})^2}{K}}$ are the mean and standard deviation of rewards of the sampled rollouts.

Finally, at each training step, a batch of (input image, label) pairs is collected from the training dataset. The above computation gives us per (input image, label) loss, we average them over all the pairs in a given batch to calculate the final loss which is then used to update the model via gradient descent.

## H.2. Training Hyperparameters

We train ResNet-50 (He et al., 2016) models on ImageNet (Deng et al., 2009) with a batch size of 256 for 20 epochs using SGD with momentum 0.9 (no Nesterov momentum) and an initial learning rate of 0.1. We swept the learning rate over $\{0.001, 0.003, 0.01, 0.03, 0.1, 0.3, 0.7, 1.0\}$ and found that the standard value of 0.1 generally works well for all objectives, so we report results at this setting. The learning rate follows a cosine schedule (Loshchilov & Hutter, 2017) with linear warmup for the first epoch. For evaluation, no augmentation is applied: each image is resized to $224 \times 224$ and normalized by the channel-wise mean and standard deviation of pixel values. During training, in addition to the same resizing and normalization, we apply a random horizontal flip (with probability 0.5) and a random resized crop to $224 \times 224$ (with scale $(0.08, 1.0)$). The number of rollouts $K$ is varied across experiments. All training runs are conducted on a single L40S GPU for 15 hours.

## H.3. Equivalence of Validation Top-1 Accuracy and Majority Voting Accuracy

In this section, we discuss the validation top-1 accuracy metric, which is the traditional metric used in image classification. Formally, validation accuracy for a single image and label pair $(x, y^*(x))$ is defined as:

$$\text{Accuracy}(x, y^*(x); \pi_\theta) = \mathbb{I}\left[\arg\max_{y \in \mathcal{Y}} \pi_\theta(y|x) = y^*(x)\right] = \begin{cases} 1, & \text{if } \arg\max_{y \in \mathcal{Y}} \pi_\theta(y|x) = y^*(x) \\ 0, & \text{otherwise} \end{cases}$$

which is then averaged over all validation examples for the final metric. In other words, validation accuracy is the same as majority voting accuracy (Wang et al., 2023a) in traditional LLM chain-of-thought reasoning tasks.

## H.4. Gradient Norm Analysis

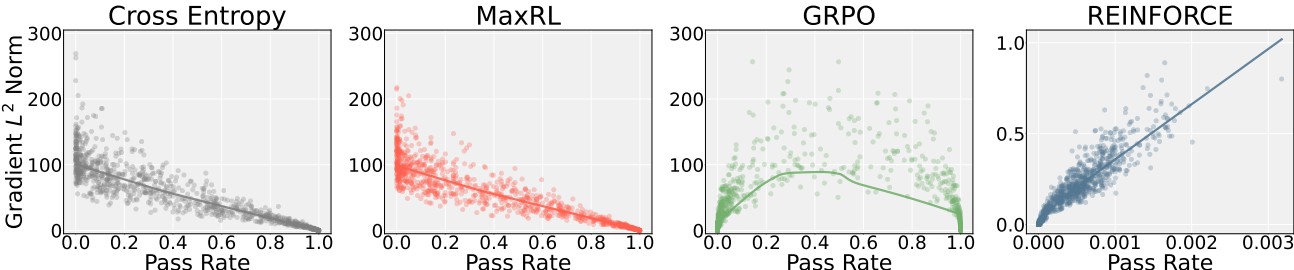

*Figure 8.* **(ImageNet gradient norm analysis)** Scatter plot, where each point has the pass rate (model's predicted probability of the correct class) of a particular image in the x-axis, and gradient $L^2$ norm for that image in the y-axis, for 1000 randomly selected images from the ImageNet validation dataset after 1500 steps of training on a ResNet-50 model. Sampling-based algorithms' (MAXRL, GRPO and REINFORCE) gradients are calculated using 131,072 rollouts per example to reduce sampling error and estimate the population-level gradient. Cross-entropy and MAXRL have similar scatter plot: with high gradient norm for hard inputs (pass rate close to 0) and lower gradient norm for the easier ones (pass rate close to 1). In contrast, the highest gradient norm for GRPO is on medium-difficulty (pass rate close to 0.5) inputs, with hard inputs having very low gradient norm. Finally, REINFORCE fails to produce any significant gradient norm and its pass rate is confined below 0.003 after 1500 steps, demonstrating its difficulty to learn in this setting.

Figure 8 shows the correlation between gradient norm and pass rate (model's predicted probability of the correct class) for a particular image on different objectives. We see that cross-entropy and MAXRL have a similar scatter plot: with high gradient norm for hard inputs (pass rate close to 0) and lower gradient norm for the easier ones (pass rate close to 1). In contrast, the highest gradient norm for GRPO is on medium-difficulty (pass rate close to 0.5) inputs, with hard inputs having very low gradient norm. Finally, REINFORCE fails to produce any significant gradient norm compared to

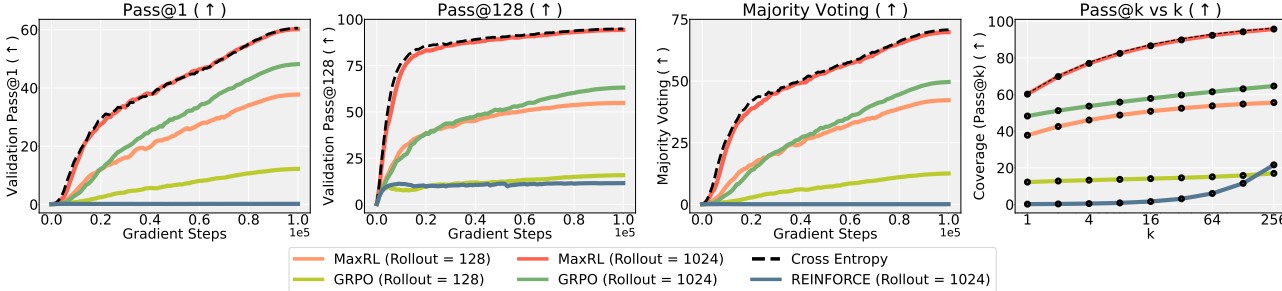

*Figure 9.* **(Additional ImageNet results)** On the didactic image classification setting, MAXRL outperforms and scales better than GRPO with additional compute, and approaches the same performance as maximum likelihood training via cross-entropy given sufficient number of rollouts ($\geq 1024$). Note that REINFORCE remains flat, since the initial model's pass rate is low ($\sim 0.1\%$) and REINFORCE fails to generate significant gradient signal (Figure 8). From left, the plots show pass@1, pass@128, Majority Voting Accuracy (equivalent to traditional validation top-1 accuracy in image classification, see Appendix H.3), and coverage of the final checkpoint, respectively.

the other objectives and its pass rate is confined below 0.003 after 1500 steps, demonstrating its difficulty to learn in this setting. This is also reflected in our other results, where REINFORCE does not show any signs of learning. We attribute this to the very low gradient norm: since the randomly initialized model has pass rate 0.001 in expectation over all inputs,

REINFORCE fails to produce sufficiently large gradients during training and therefore stalls in model improvement. One caveat: REINFORCE's failure may be due to us training the model from scratch — on a pretrained model, it indeed produces gradients but still shows poor gradient norm on hard inputs (see Figure 6).

### H.5. Comparison with GRPO and Additional Metrics

Here we present additional experimental results. In particular, **(1)** we compare against GRPO with varying number of rollouts, **(2)** record additional metrics such as majority voting accuracy (i.e., validation top-1 accuracy), and **(3)** show the resulting coverage (pass@k vs k) from different objectives. Figure 9 records our findings: MAXRL outperforms and scales better than GRPO with additional compute. While GRPO improves performance if given more compute unlike REINFORCE, it remains suboptimal compared to MAXRL and supervised cross-entropy training. Moreover, both GRPO and REINFORCE exhibit worse coverage as their pass@k values are significantly lower compared to MAXRL, corroborating our experiments from other sections.

# I. Maze Experiments

## I.1. Model Architecture

We adopt a lightweight decoder-only Transformer model following the Qwen2 architecture (Yang et al., 2024a), with a total of approximately $3M$ parameters. The model consists of 4 Transformer layers, each using full self-attention. The hidden size is set to 256, with an intermediate (feed-forward) dimension of 1024, and 4 attention heads per layer. We use grouped query attention with 2 key-value heads. The model employs RMSNorm with $\sigma = 1 \times 10^{-6}$ and uses the SiLU activation function in the feed-forward networks. Rotary positional embeddings (RoPE) (Su et al., 2024) are applied with $\theta = 1,000,000$, and the maximum sequence length is 512 tokens. The vocabulary size is 32 tokens, and input and output embeddings are tied. The model is trained and evaluated using bfloat16 precision, with attention dropout set to 0. The architecture follows a standard causal language modeling setup with autoregressive decoding.

## I.2. Task Description

Mazes are procedurally generated using Prim's algorithm (Prim, 1957), and task difficulty is controlled by the grid size. We use a symbolic tokenization to represent both the maze layout and the navigation policy, with tokens drawn from a small, discrete vocabulary.

The input sequence describes a two-dimensional grid in row-major order. Each cell is represented by a single token indicating its type (e.g., WALL, PATH, START, or GOAL). Rows are separated by a dedicated NEWLINE token, and the entire grid is delimited by special boundary tokens marking the beginning (GRID_START) and end (GRID_END) of the grid description. Following the maze specification, the model autoregressively generates a sequence of navigation actions drawn from a fixed action vocabulary (e.g., directional moves) and terminates by a DONE token.

Below, we provide an example data instance following this format.

---

**7*7 Maze Example Model Input and Output Format**

**Input:**
```
<bos> GRID_START WALL WALL WALL WALL WALL WALL WALL NEWLINE WALL START WALL PATH PATH
PATH WALL NEWLINE WALL PATH WALL PATH WALL WALL WALL NEWLINE WALL PATH PATH PATH PATH
PATH WALL NEWLINE WALL PATH WALL WALL WALL PATH WALL NEWLINE WALL PATH WALL PATH PATH
GOAL WALL NEWLINE WALL WALL WALL WALL WALL WALL WALL NEWLINE GRID_END PATH_START
```
**Output:**
```
RIGHT RIGHT RIGHT RIGHT DOWN DOWN DOWN DOWN DONE <eos>
```

---

For reference, we also visualize one typical successful trajectory and one representative failed prediction in Figure 10.

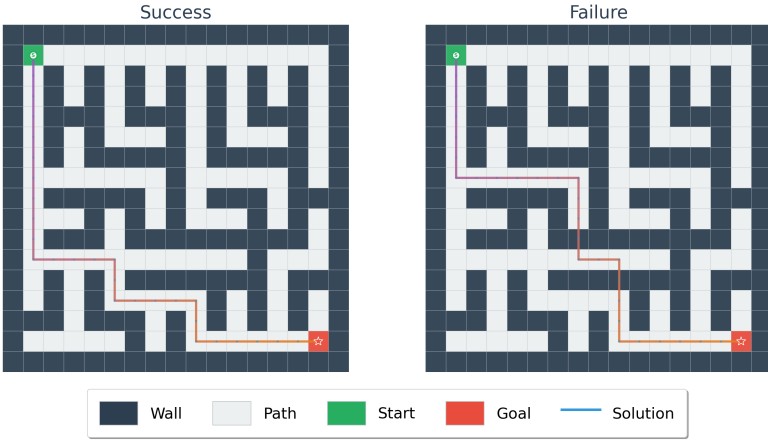

*Figure 10.* **(Maze data visualization)** The left plot shows a successful navigation trajectory, while the right plot illustrates a failure case produced by the trained model, where the generated action sequence deviates from the correct path before reaching the goal.

### I.3. Training Setups

To ensure sufficient task complexity and rigorous evaluation, we construct a training set of 1 million distinct $17 \times 17$ mazes and a test set of 256 non-overlapping samples. We first pretrain the model from scratch, where it is trained to follow a provided ground-truth trajectory for each maze. During SFT, we use a learning rate of $5 \times 10^{-4}$ with the AdamW optimizer (Kingma & Ba, 2015; Loshchilov & Hutter, 2019) and train for 1,500 steps with a batch size of 32. This pretraining stage[3] initializes the model with the basic output format required for representing maze-solving trajectories.

Subsequently, we perform reinforcement learning (RL) training, where we use a learning rate of $1 \times 10^{-4}$ in all our experiments. We update the model parameters only once per RL step (i.e., the fully on-policy setting from Tajwar et al. (2024)). We have two different training settings:

1. **256 Batch size, 9000 RL steps, varying number of training rollouts.** This is used for Figures 3 and 11. Figure 3 compares performance across different number of training rollouts (from 4 to 128). We observed that when number of training rollouts is sufficiently small (e.g., 4), lower batch size training runs can collapse, possibly due to noise/gradient variance. Hence these runs use a larger batch size of 256. Since this setting is very expensive, we only train our models for 9000 RL steps, and only train with MAXRL and our principal baselines, namely GRPO and RLOO. Figure 11 shows the evolution of various Pass@k metrics under these 3 objectives during training, and reports experiments resulting from our highest compute setting, namely that with 128 training rollouts per prompt.

2. **32 Batch size, 20000 RL steps, 128 training rollouts per task.** This setting is used for Figure 12, where we show the training curves for different baselines on the maze setting, and Table 3, where we present summary Pass@k performance of different baselines at the end of 20000 training steps. Our first setting of 256 batch size was beyond our compute budget for being too expensive to train all the different baselines with, so we adopted this cheaper setting for baseline comparison.

Each training run uses 4 RTX 4090 GPUs. Given the small number of model parameters, the model is no longer memory-bound, so we modified the rollout engine (instead of using the default vLLM (Kwon et al., 2023) engine) in the verl (Zhang et al., 2024; Sheng et al., 2024) codebase to increase sampling parallelism and speed up training.

### I.4. Training Dynamics in Our Highest Compute Budget Setting

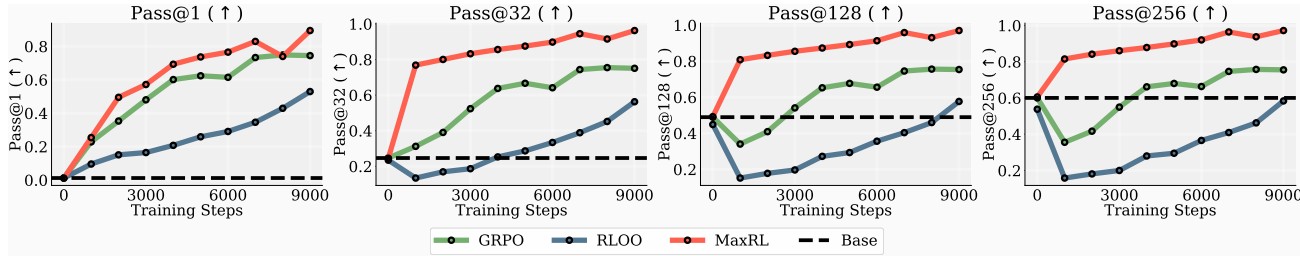

*Figure 11.* (**Training dynamics in the maze setting under our highest compute budget**) We investigate how the Pass@k performance of different objectives evolve when we train a 3M model to solve 17x17 maze puzzles on our highest compute budget, namely 256 batch size and 128 training rollouts per task. MAXRL performs significantly better compared with GRPO and REINFORCE in pass@1, pass@32, pass@128 and pass@256. These results signify MAXRL's effectiveness in compute scaling during RL.

Here we provide additional results pertaining to the training dynamics of MAXRL in the infinite data maze setting. Figure 11 shows pass@1 through pass@256 over training steps for MAXRL, GRPO, and RLOO, trained for 9000 RL steps using 256 batch size and 128 training rollouts per task. MAXRL outperforms both GRPO and RLOO at all metrics.

---

[3]Our pretrained Maze transformer checkpoint, which is used as the starting model for RL fine-tuning, can be found here: https://github.com/tajwarfahim/maxrl/tree/main/maze/ckpt-1500.

## I.5. Comparison Table with Additional Baselines

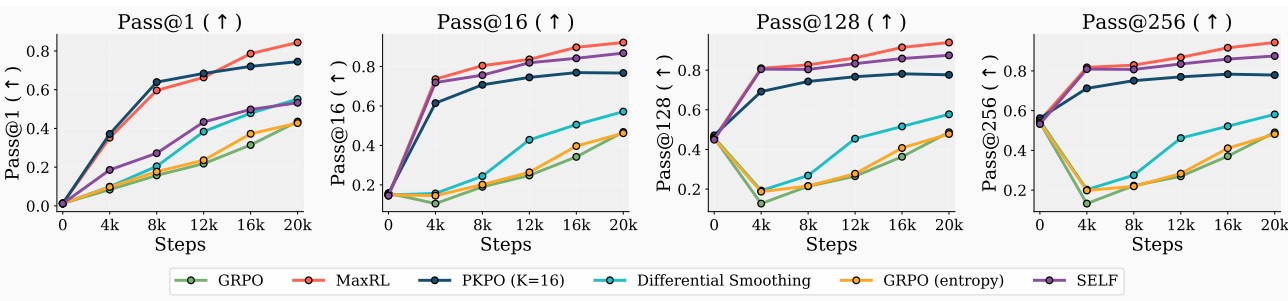

*Figure 12.* (**Training curves compared with other baselines.**) We compare MAXRL with other RL algorithms, including entropy bonus, PKPO (Walder & Karkhanis, 2025), Differential Smoothing (Gai et al., 2026) and SELF (Nguyen et al., 2025b), under batch size 32 and number of training rollouts 128. MAXRL significantly outperforms other methods in all metrics, and is the only method to maintain both good average performance (pass@1) and coverage (pass@k).

*Table 3.* Performance comparison across methods in maze (batch size 32, number of training rollouts per task 128).

| Method | Pass@1 | Pass@16 | Pass@128 | Pass@256 |
|---|---|---|---|---|
| GRPO | 39.6 | 41.3 | 42.4 | 43.0 |
| GRPO (with entropy bonus) | 47.2 | 51.7 | 53.4 | 54.0 |
| PKPO (T = 16) (Walder & Karkhanis, 2025) | 74.5 | 76.7 | 77.6 | 77.9 |
| SELF (Nguyen et al., 2025b) | 46.1 | 82.7 | 86.3 | 87.5 |
| Differential Smoothing (Gai et al., 2026) | 50.0 | 57.1 | 57.8 | 58.7 |
| **MAXRL** | **84.4** | **92.2** | **94.0** | **94.3** |

Table 3 compares MAXRL against a broader set of baselines in the Maze setting, where all models are trained for 20,000 steps with a batch size of 32 and 128 rollouts per prompt. MAXRL reaches 84.4 pass@1 and 94.3 pass@256, improving over standard GRPO by more than 44 points on pass@1. PKPO attains a strong pass@1 of 74.5 but its pass@k curve is nearly flat, indicating that it has largely collapsed onto a single mode and gains little from additional samples. SELF reaches a high pass@256 of 87.5 yet a weak pass@1 of 46.1, trading away average accuracy for coverage. GRPO, with or without an entropy bonus, and Differential Smoothing all remain well below 60 across the board; among these, Differential Smoothing performs slightly better than GRPO with an entropy bonus, which in turn improves over vanilla GRPO. Figure 12 shows the evolution of the Pass@k metrics for these baselines throughout during. Overall, MAXRL shows superior performance on both average accuracy (pass@1) and coverage (pass@k).

# J. GSM8K Experiments

## J.1. Prompt template

The goal of this experiment setting is to study how RL algorithms perform in data-scarce but compute rich settings. This is similar in spirit to Kim et al. (2026), except that we study RL and not pre-training. Due to our compute limitations, we choose a relatively small model (360M parameters) for our experiments. Specifically, we conduct reinforcement finetuning on the `SmolLM2-360M-Instruct` model (Allal et al., 2025) using the GSM8K (Cobbe et al., 2021) training set for 1500 steps. We use GSM8K-Platinum (Vendrow et al., 2025), a revised version of the original GSM8K test that is non-overlapping with the GSM8K train dataset, for evaluation. An example prompt and the model's response are shown below. In particular, we take each individual problem, append it with " Let's think step by step and output the final answer within \\boxed{}." and process the string through the SmolLM2-360M-Instruct model's tokenizer chat-template to obtain the final prompt for the model.

---

**SmolLM2-360M-Instruct**

```
<|im_start|>system
You are a helpful AI assistant named SmolLM, trained by Hugging Face<|im_end|>
<|im_start|>user
Jeannie hikes the 12 miles to Mount Overlook at a pace of 4 miles per hour, and then
returns at a pace of 6 miles per hour.  How long did her hike take, in hours?  Let's
think step by step and output the final answer within \\boxed{}.  <|im_end|>
<|im_start|>assistant
To find the time it takes for Jeannie to hike the entire trip, we need to calculate
her total travel time and then divide it by the rate at which she travels.
Jeannie hikes the 12 miles at a pace of 4 miles per hour.  So, the time taken is 12
miles / 4 miles per hour = 3 hours.
Next, she returns at a pace of 6 miles per hour.  So, the time taken to return is 12
miles / 6 miles per hour = 2 hours.
Therefore, the total time taken is 3 hours + 2 hours = 5 hours.
So, the final answer is 5 hours.  \\boxed{5} <|im_end|>
```

---

## J.2. Hyperparameters

Detailed hyperparameter setup is provided in Table 4. For other baselines, we use $T = 16$ for PKPO, which means we optimize pass@16 as the objective, and $\gamma = 0.05$ for differential smoothing.

*Table 4.* Training hyperparameters for SmolLM2-360M-Instruct on GSM8K.

| Training hyperparameters for SmolLM2-360M-Instruct | | | |
|---|---|---|---|
| **Parameter** | **Value** | **Parameter** | **Value** |
| Base model | SmolLM2-360M-Instruct | Training set | GSM8K |
| Test set | GSM8K | Prompts per batch | 256 |
| Generations per prompt | 128 | Grad update per RL step | 1 |
| Max prompt length | 512 | Max response len | 2048 |
| Learning rate | $1 \times 10^{-5}$ | Training Steps | 1500 |
| KL coeff | 0.0 | Entropy coeff | 0.0 |
| Rollout temp | 1.0 | Validation top_p | 0.95 |
| Validation temp | 0.6 | Device | $8 \times$ Nvidia GH200 |

## J.3. Effect of Additional Compute

In Figure 2, we demonstrated how MAXRL's performance scales as additional compute is available in terms of rollout budget, but in our supervised ImageNet classification setting. Here, we further experiment with the effect of additional

compute on the finite data GSM8K training setting, which is a more realistic RL training since it involves autoregressive generations from a 360M LLM on a real mathematical reasoning task.

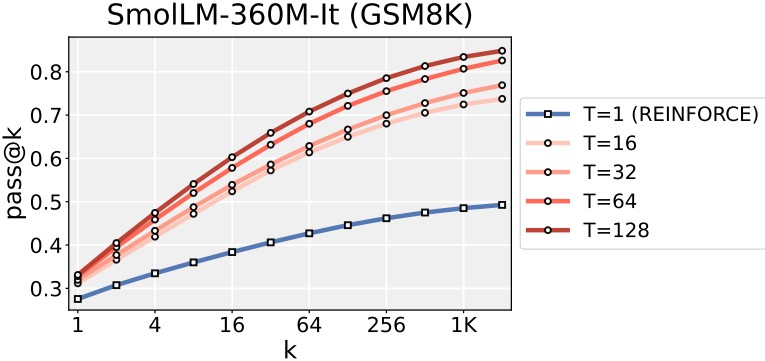

*Figure 13.* (**Effect of number of training rollouts in the GSM8K setting**) For MAXRL, increase in number of rollouts $N$ directly leads to an increase in the approximation order $T$ of the maximum likelihood objective that the finite sample MAXRL optimizes. Here we compare the final checkpoint resulting from training with different values of $T(=$ Number of rollouts) but otherwise identical training setup on the GSM8K setting. Increasing compute smoothly translates into performance gain, with the highest amount of gain for higher pass@k.

Figure 13 shows our findings: as the number of training rollouts increases, it results in performance improvement in the final checkpoint. Moreover, as expected, the biggest gains come at higher values of pass@k, with $T = 128$ outperforming $T = 1$ (REINFORCE) by almost $30\%$ at pass@2048.

### J.4. Comparison with Additional Baselines

*Table 5.* Performance comparison across methods on GSM8K.

| Method | Pass@1 | Pass@16 | Pass@128 | Pass@1024 |
|---|---|---|---|---|
| GRPO (Shao et al., 2024) | 29.3 | 40.1 | 45.8 | 48.8 |
| RLOO (Ahmadian et al., 2024) | 27.5 | 38.8 | 44.6 | 48.5 |
| GRPO (with entropy bonus) | 31.1 | 42.4 | 48.1 | 51.6 |
| PKPO (T = 16) (Walder & Karkhanis, 2025) | 30.7 | 53.6 | 67.2 | 75.9 |
| Differential Smoothing (Gai et al., 2026) | 31.4 | 42.7 | 48.5 | 52.3 |
| **MAXRL** | **33.2** | **60.3** | **75.0** | **83.4** |

Table 5 compares MAXRL against the same set of baselines in the data-scarce GSM8K training setting (Table 4). MAXRL again achieves the best performance on every metric, reaching 33.2 pass@1 and 83.4 pass@1024. While all methods are clustered within a few points at pass@1 (between 27.5 and 33.2), the differences widen dramatically as $k$ grows: MAXRL attains 83.4 pass@1024, 7.5 points above the strongest baseline (PKPO at 75.9) and more than 30 points above GRPO, RLOO, GRPO with an entropy bonus, and Differential Smoothing, all of which saturate around 49–52. These results suggest MAXRL extracts more learning signal from a fixed training dataset.

### J.5. Per-Prompt Pass-Rate Distribution During Training

To visualize the sharpening dynamics of the three methods, Figure 14 tracks the empirical distribution of per-prompt pass rates over the GSM8K training set throughout the SmolLM2-360M-Instruct run from Table 4. Each row corresponds to a training epoch $(0, 10, 20, 30, 40, 50)$, each column to a method, and each bar to the fraction of training prompts whose pass-rate estimate falls into the corresponding $\log_2$-spaced bin (with the leftmost bin reserved for prompts with 0 correct rollouts and the rightmost for prompts solved on every rollout). Validation pass@1 values of the corresponding checkpoints over the entire evaluation dataset are reported on top of each panel.

Figure 14 suggests that GRPO and RLOO push the per-prompt pass-rate distribution towards either 0 or 1, supporting the distribution sharpening narrative, whereas MAXRL retains a broad distribution of pass rates throughout training. This

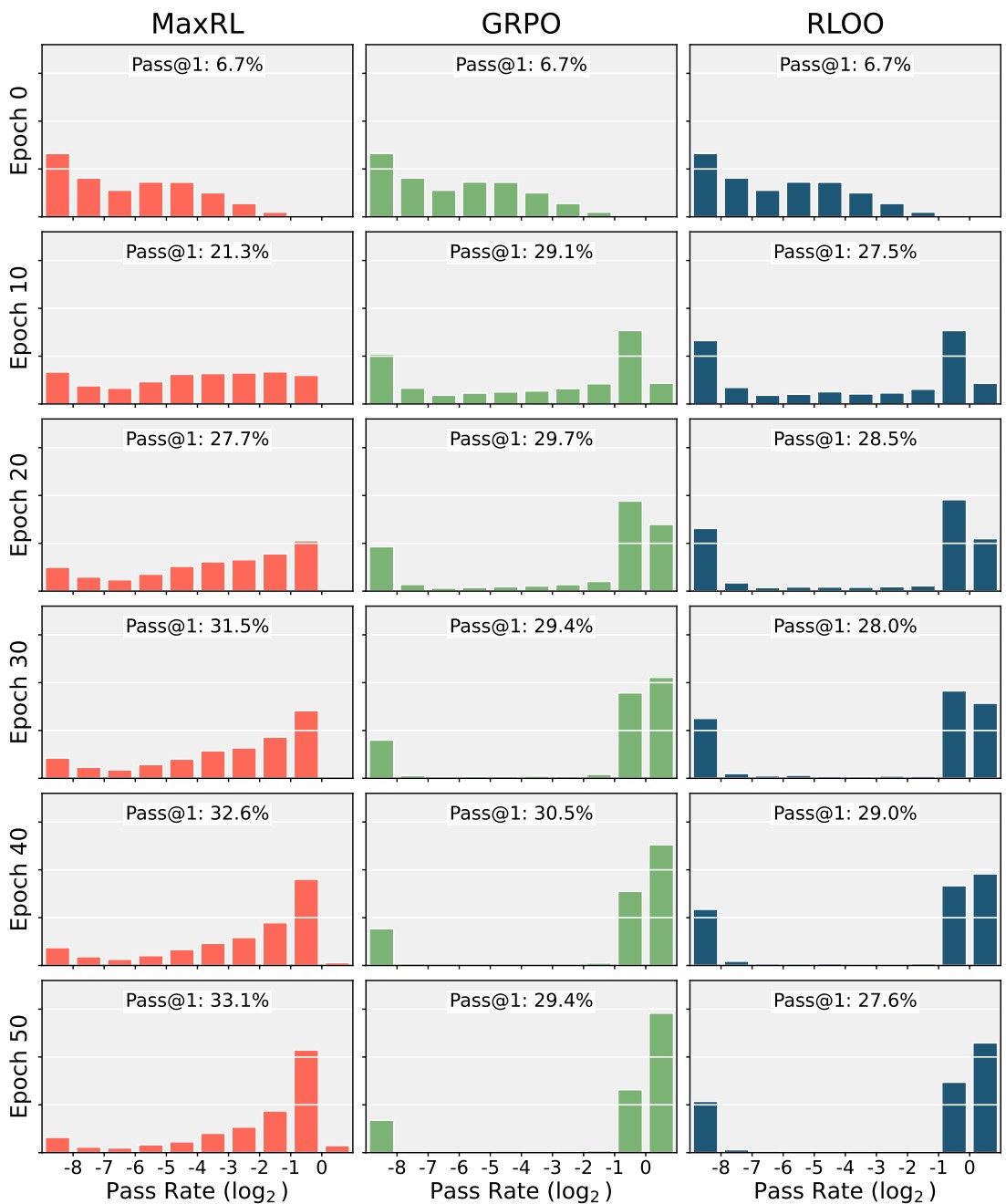

*Figure 14.* (**Distribution sharpening over training on GSM8K**) Evolution of the per-prompt pass-rate distribution on the SmolLM2-360M-Instruct GSM8K training set across 50 training epochs for MAXRL, GRPO, and RLOO. All three methods start from the same base-model distribution (epoch 0). As training progresses, GRPO and RLOO concentrate prompts into the two extreme bins ($[0, 0]$ and $[1, 1]$), so the model either trivially solves or trivially fails most prompts and the intermediate bins are emptied. In contrast, MAXRL keeps probability mass spread across the entire pass-rate spectrum, gradually shifting weight from the failure end toward the higher pass-rate bins while continuing to expose intermediate-difficulty prompts to non-zero learning signal.

contrast shows that MAXRL is more robust to the distribution sharpening commonly observed in LLM reinforcement learning (Yue et al., 2025; Dang et al., 2025; Wu et al., 2026), allowing it to extract more learning signal from a fixed training dataset.

# K. Qwen3 Experiments

## K.1. Prompt template

We use the Qwen-math template (Yang et al., 2024a; Qwen et al., 2025; Yang et al., 2024b) for formatting our prompts. We show an example prompt (Yu et al., 2025) after formatting through our template below. In particular, we take each individual problem, append it with "\nPlease reason step by step, and put your final answer within \\boxed{{}}." and process the string through the Qwen3 model's tokenizer chat-template to obtain the final prompt for the model.

---

**Qwen Math Prompt Template**

```
<|im_start|>system
Please reason step by step and put the final answer in \\boxed{}.  <|im_end|>
<|im_start|>user
Denote by S(n) the sum of the digits of the positive integer n.  Find all the solutions
of the equation n(S(n) − 1) = 2010.  Let's think step by step and output the final answer
within \\boxed{}.  <|im_end|>
<|im_start|>assistant
```

---

## K.2. Hyperparameters

Next, we describe the default hyperparameters for our training setup. Since there are many possible alternatives to handle off-policy updates and corresponding importance ratio (Schulman et al., 2017; Shao et al., 2024; Zheng et al., 2025; MiniMax et al., 2025; Yu et al., 2025), to keep things simple, we choose to train in the fully on-policy setup, meaning we have no importance ratio or associated clipping. Similarly, to avoid tuning additional hyperparameters for each algorithm, following Olmo et al. (2025), we remove KL penalty and also entropy bonus in our default training comparison. Note: we train with GRPO and entropy bonus as a baseline in our SmolLM2-360M-Instruct training on GSM8K, results are recorded in Table 5: MAXRL outperforms this variant, showing that entropy bonus does not fully mitigate issues resulting from GRPO though it can slightly mitigate it, as also observed by Yue et al. (2025).

We generate all training rollouts using temperature 1.0, and do not use special sampling techniques. Similarly, we also do not use any adaptive sampling (Yu et al., 2025) or fixes for inference-training logit mismatch (He, 2025; Khatri et al., 2026). Finally, for evaluation, we follow the same protocol as Yue et al. (2025), and we run inference with temperature 0.6, top-p sampling parameter 0.95, no top-k or min-p sampling (Nguyen et al., 2025a).

Table 6 shows our default hyperparameter setting.

*Table 6.* Training hyperparameters for Qwen3-1.7B-Base and Qwen3-4B-Base training.

| Training hyperparameters for Qwen3-1.7B-Base and Qwen3-4B-Base | | | |
|---|---|---|---|
| **Parameter** | **Value** | **Parameter** | **Value** |
| Base model | Qwen3-1.7B-Base, Qwen3-4B-Base | Prompts per batch | 256 |
| Generations per prompt | 16 | Grad update per RL step | 1 |
| Max prompt length | 1024 | Max response len | 4096 |
| Learning rate | $1 \times 10^{-6}$ | Training Steps | 1000 |
| KL coeff | 0.0 | Entropy coeff | 0.0 |
| Rollout temp | 1.0 | Validation top-p | 0.95 |
| Validation temp | 0.6 | Device | $32 \times$ Nvidia H200 |

## K.3. Additional Results

We report additional empirical results from our Qwen3-1.7B-Base and Qwen3-4B-Base training runs that complement the main-paper evaluation. We expand the benchmark coverage with four further reasoning datasets, compare MAXRL against the base model and GRPO under verifier-free majority voting, and report training-dynamics traces (response length, actor entropy, gradient norm) as well as validation pass@1 curves throughout training.

**Extended Benchmarks.** Beyond the four primary benchmarks reported in the main paper, we further evaluate the Qwen3-4B-Base trained model on four additional reasoning benchmarks: AIME 2024, HMMT Feb 2025 (MathArena, 2025a), HMMT Nov 2025 (MathArena, 2025b), and JEEBENCH (Arora et al., 2023). Figure 15 reports coverage on these benchmarks; MAXRL matches or outperforms both the base model and GRPO, achieving up to $20.5\times$ speedup over GRPO when generating multiple samples under a perfect verifier while maintaining similar or better pass@1 performance.

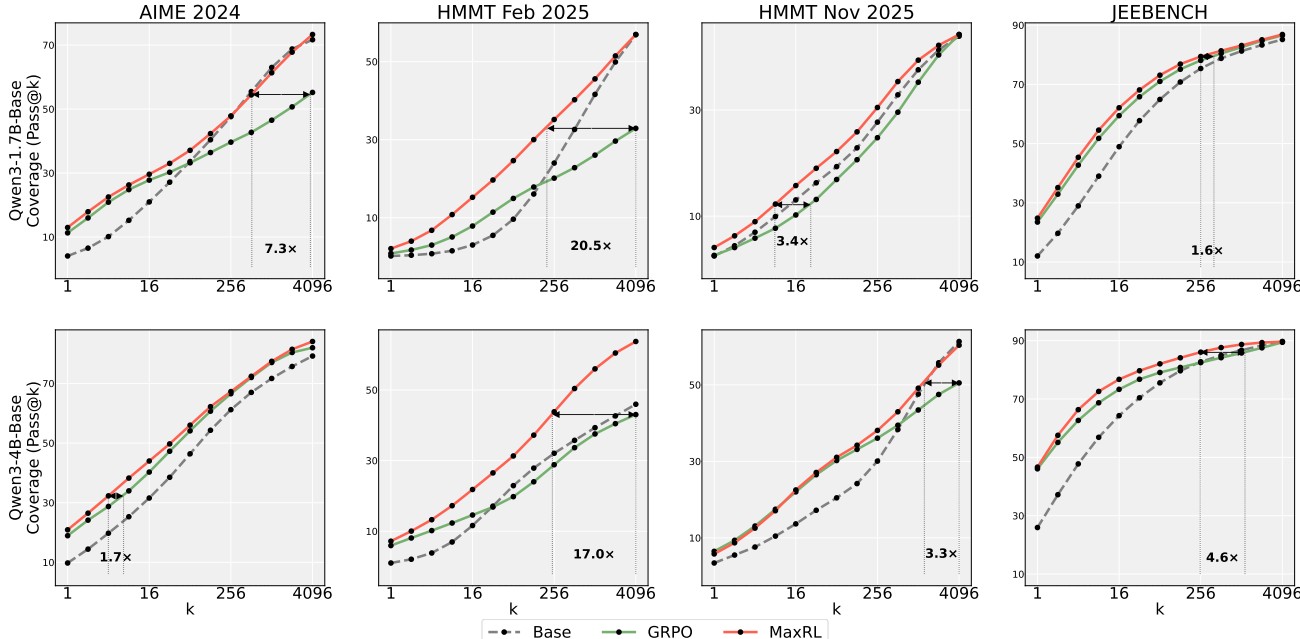

*Figure 15.* (**Evaluation of Qwen3 model training on additional benchmarks**) Coverage on AIME 2024, HMMT Feb 2025 (MathArena, 2025a), HMMT Nov 2025 (MathArena, 2025b), and JEEBENCH (Arora et al., 2023). MAXRL matches or outperforms both the base model and GRPO, leading to up to $20.5\times$ speedup compared to GRPO while generating multiple samples with a perfect verifier and maintaining similar or better pass@1 performance.

**Majority voting.** We compare MAXRL against the base model and GRPO under *majority voting* (Wang et al., 2023a), a widely used verifier-free method for scaling test-time compute: we draw $N$ i.i.d. rollouts for each prompt, group the responses by their final answer, and take the most frequent answer as our prediction. Table 7 reports majority-voting accuracy on five benchmarks; MAXRL outperforms both the pretrained base model and the GRPO-trained model on every benchmark.

*Table 7.* (**Majority voting performance comparison on Qwen3-4B-Base**) We compare the performance of MAXRL in terms of majority voting against the pretrained base model and GRPO.

|  | AIME 2024 (majority@4096) | AIME 2025 (majority@4096) | BeyondAIME (majority@4096) | MATH-500 (majority@2048) | Minerva (majority@2048) |
|---|---|---|---|---|---|
| Base | 23.3 | 23.3 | 7.0 | 69.8 | 18.8 |
| GRPO | 23.3 | 23.3 | 7.0 | 72.4 | 27.2 |
| MAXRL | **26.7** | **26.7** | **14.0** | **74.0** | **28.7** |

**Training dynamics.** We record mean generated response length, actor entropy, and gradient norm throughout training for both Qwen3-4B-Base (Figure 16) and Qwen3-1.7B-Base (Figure 17). The trends are consistent across model sizes: MAXRL produces longer chains-of-thought, retains higher actor entropy, and exhibits larger gradient norms during training compared with GRPO.

**Validation accuracy during training.** In addition, we examine pass@1 (estimated using 32 samples) over the course of training on intermediate evaluation datasets. For Qwen3-4B-Base, MAXRL consistently outperforms GRPO across the three evaluation datasets (Figure 18). For the smaller Qwen3-1.7B-Base model (Figure 19), MAXRL initially trails GRPO

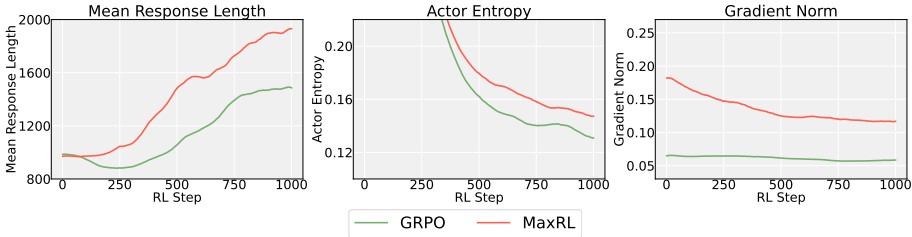

*Figure 16.* (**Additional training dynamics metrics for Qwen3-4B-Base**) We show comparison between GRPO and MAXRL in terms of mean response length, entropy of the actor, and gradient norm during training for the Qwen3-4B-Base model. MAXRL generally produces longer chains-of-thought, and also retains higher actor entropy during training. MAXRL also produces larger gradient norms during training.

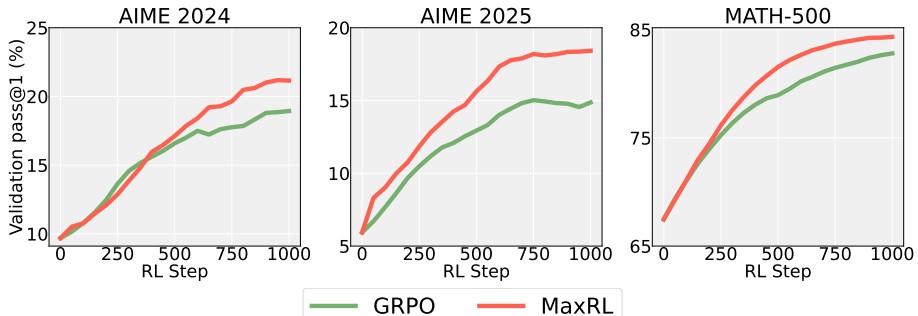

*Figure 17.* (**Additional training dynamics metrics for Qwen3-1.7B-Base**) We show comparison between GRPO and MAXRL in terms of mean response length, entropy of the actor, and gradient norm during training for the Qwen3-1.7B-Base model. MAXRL generally produces longer chains-of-thought, and also retains higher actor entropy during training. MAXRL also produces larger gradient norms during training.

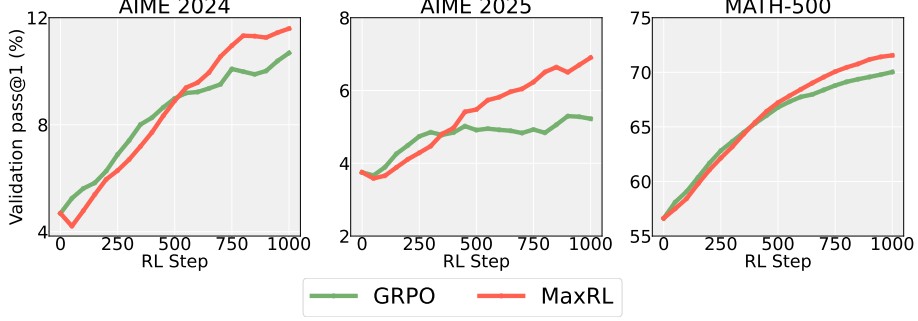

*Figure 18.* (**Qwen3-4B-Base validation pass@1 during training**) Pass@1 (estimated using 32 samples) during training of Qwen3-4B-Base, on 3 different evaluation datasets. MAXRL consistently outperforms GRPO during training.

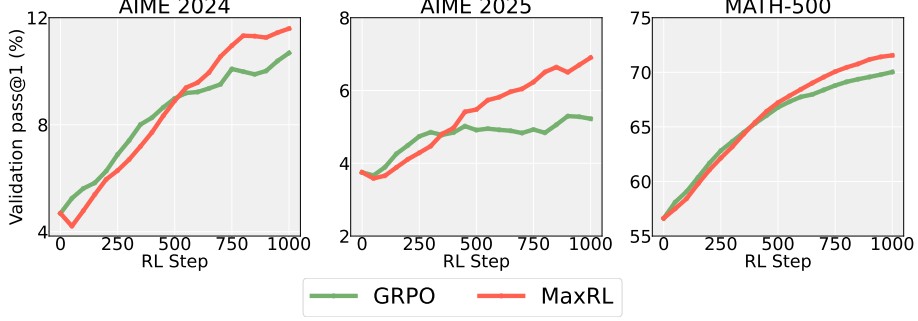

*Figure 19.* (**Qwen3-1.7B-Base validation accuracy during intermediate training**) We record validation pass@1 (using mean over 32 rollouts per prompt) over AIME 2024, AIME 2025 and MATH-500 during Qwen3-1.7B-Base model training.

at pass@1, similar to the SmolLM curve in Figure 4, but catches up with extended training and ultimately converges to a higher value.

# L. Comparison with Additional Baselines

## L.1. PKPO

Realizing the shortcomings of the expected reward (pass@1) optimization paradigm in reinforcement learning, recent work (Walder & Karkhanis, 2025; Tang et al., 2025; Bagirov et al., 2025) has also proposed directly optimizing for pass@k during RL training. The connection between maximum likelihood and harmonic mixture of pass@k optimization necessitates a direct comparison between our work and these pass@k optimization procedures. Here we present comparisons with a representative baseline, PKPO (Walder & Karkhanis, 2025). PKPO has a tunable hyperparameter $T$, which dictates the order of optimization objective, namely PKPO with parameter $T$ optimizes pass@$T$ directly.

*Table 8.* Comparison between PKPO (T = 16) and MAXRL on the maze setting.

| Method | Pass@1 | Pass@16 | Pass@128 | Pass@256 |
|---|---|---|---|---|
| PKPO (T = 16) (Walder & Karkhanis, 2025) | 74.5 | 76.7 | 77.6 | 77.9 |
| **MAXRL** | **84.4** | **92.2** | **94.0** | **94.3** |

*Table 9.* Comparison between PKPO (T = 16) and MAXRL on the GSM8K setting.

| Method | Pass@1 | Pass@16 | Pass@128 | Pass@1024 |
|---|---|---|---|---|
| PKPO (T = 16) (Walder & Karkhanis, 2025) | 30.7 | 53.6 | 67.2 | 75.9 |
| **MAXRL** | **33.2** | **60.3** | **75.0** | **83.4** |

Tables 8 and 9 show comparison between PKPO and MAXRL on the maze and GSM8K settings, respectively. PKPO experiments are run with $T = 16$, with all other hyperparameters kept the same as their default values for the respective settings. We observe that MAXRL outperforms PKPO at all values of pass@k. **More importantly, MAXRL outperforms PKPO at pass@16, despite the latter explicitly optimizing for this metric.** A similar phenomenon is observed with REINFORCE, which directly optimizes pass@1 yet consistently underperforms MAXRL at pass@1. This suggests that directly optimizing a specific pass@k objective does not necessarily yield the best performance at that pass@k. We hypothesize that MAXRL's aggregation of signals across pass@k events enables more effective use of the learning signal, and leave a detailed investigation to future work.

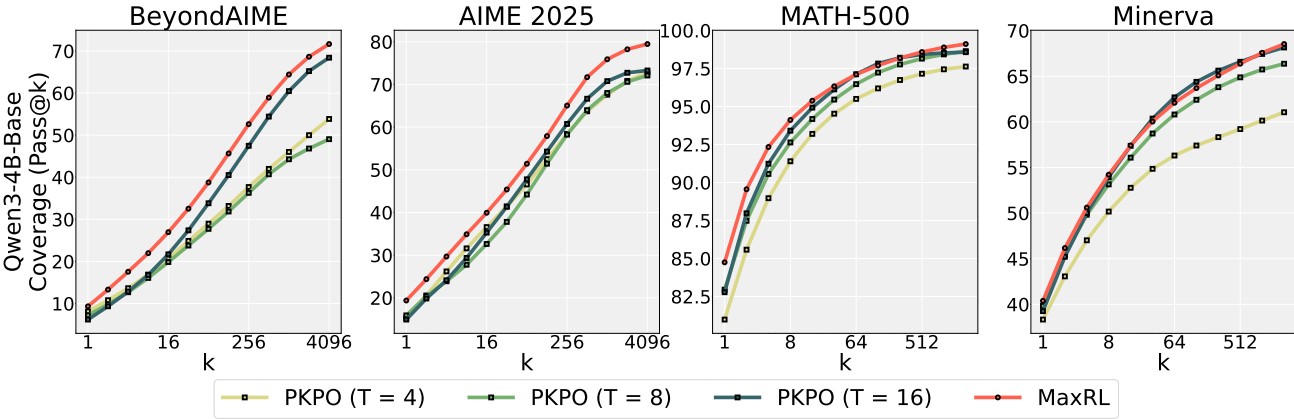

*Figure 20.* (**Comparison with PKPO on Qwen3-4B-Base training**) We compare MAXRL against PKPO for $T \in \{4, 8, 16\}$. MAXRL retains the same performance or outperforms PKPO with different values for $T$. On harder evaluation datasets like BeyondAIME and AIME 2025, the gap is more pronounced, and MAXRL outperforms PKPO at pass@k for all values of k that we evaluated on.

Figure 20 shows further comparisons against PKPO on the large scale LLM training setting. Specifically, we train a Qwen3-4B-Base model on the POLARIS-53K dataset, using the default hyperparameters for this setting. For PKPO, we train with 3 different values of $T$, namely 4, 8, and 16. MAXRL retains similar performance or outperforms PKPO on all 4 evaluation datasets. On harder evaluation datasets like BeyondAIME and AIME 2025, the gap is more pronounced, and MAXRL outperforms PKPO at pass@k for all values of k that we evaluated on. These large-scale training experiments validate our results from the smaller maze and GSM8K settings.

### L.2. ZPD

One way of thinking about MAXRL is to focus on its upweighting of high-difficulty prompts. This raises the question: how does MAXRL compare to methods that perform difficulty-aware sampling strategies? RLOO with a "not too easy, not too hard" curriculum (ZPD (Bae et al., 2026)) can be interpreted as reweighting prompts with weight $w(p) = p(1 - p)$, where $p = p_\theta(x)$ is the pass rate for prompt $x$, in Table 2. Since we do not use any additional curriculum learning or involved sampling strategies in our setting for the sake of simplicity, we choose to compare against this variant of ZPD by employing the advantage function $\hat{A}(x, y) = \hat{r}(1 - \hat{r})(r(x, y) - \hat{r})$, where $\hat{r}$ is the Monte-Carlo estimate of the pass rate for prompt $x$.

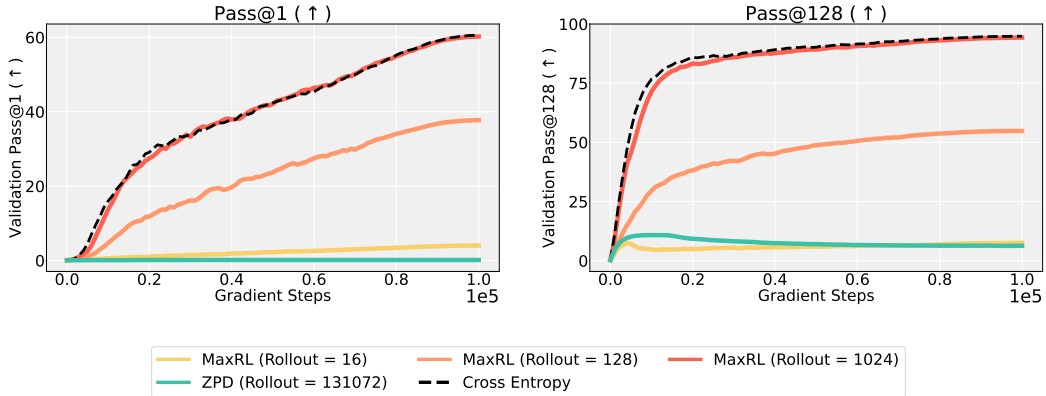

*Figure 21.* (**Comparison with ZPD on supervised ImageNet classification setting**) We compare MAXRL against ZPD on the ImageNet setting under identical training setup. Similar to REINFORCE, ZPD fails to improve significantly beyond the randomly initialized model.

First, we compare against ZPD on the ImageNet classification setting in Figure 21. ZPD performs similarly to REINFORCE here, and fails to improve the model significantly beyond its random initialization at the beginning of training.

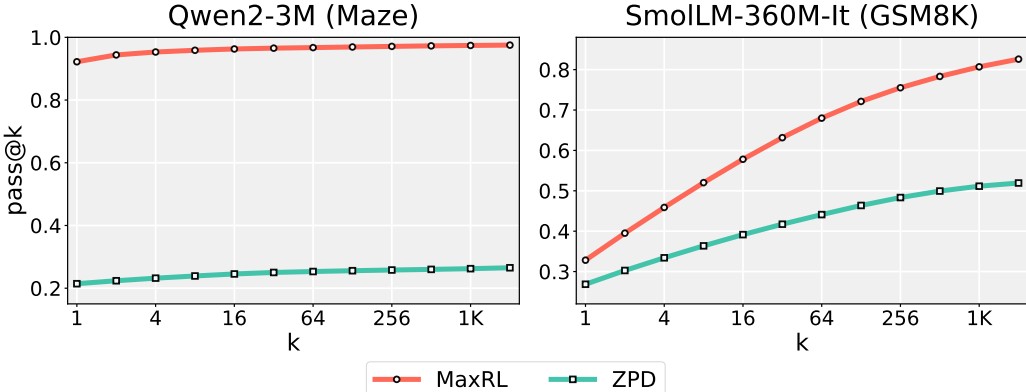

*Figure 22.* (**Comparison with ZPD in the maze and GSM8K setting.**) We compare MAXRL against ZPD on the infinite data maze and finite data GSM8K training settings under identical training budget and hyperparameters. On both settings, MAXRL significantly outperforms ZPD at pass@k, for all values of k that we evaluated on.

Next, we evaluate ZPD on the maze and GSM8K training settings. Figure 22 shows our main results: under identical training settings and compute budget, MAXRL outperforms ZPD on both settings, achieving significantly higher pass@k for all values of k that we evaluated on. On GSM8K, MAXRL achieves $\sim 30\%$ improvement in pass@2048 over ZPD.

Finally, we also compared MAXRL with ZPD on our largest training setting: Qwen3-4B-Base trained on POLARIS-53K. Figure 23 shows our main findings: under identical training setup and compute budget (1000 steps of RL training), MAXRL outperforms ZPD on all evaluation datasets, achieving gains of $12.6\times$ to $29.0\times$ in test-time scaling efficiency under a perfect verifier.

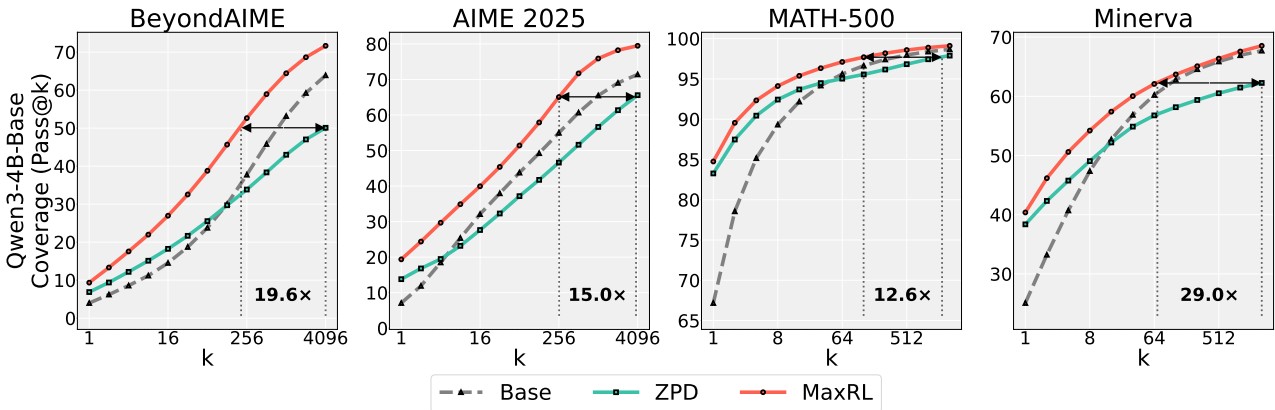

*Figure 23.* (**Comparison with ZPD on Qwen3-4B-Base training**) We compare MAXRL against ZPD while training a Qwen3-4B-Base model on the POLARIS-53K dataset under identical training setup and compute budget (1000 steps of RL training). MAXRL outperforms ZPD on evaluation datasets, achieving 12.6× to 29.0× gains in test-time scaling efficiency under a perfect verifier.

### L.3. Power Function

Finally, MAXRL can be seen as optimizing a particular non-linear function of the pass rate ($\log p_\theta(x)$). This begs the question of how other non-linear functions of the pass rate may perform. Here we compare with an example function from Xiong et al. (2025b), namely, the Power Function weighting. The Power Function with parameter $\alpha > 0$ is defined as $f(t) = t^\alpha$, and correspondingly one can optimize $J^\alpha_{\text{power-function}} = \mathbb{E}_{x \sim \rho}[p_\theta(x)]^\alpha$. The corresponding gradient of this objective is:

$$\nabla_\theta J^\alpha_{\text{power-function}} = \mathbb{E}_{x \sim \rho}\left[\alpha p_\theta(x)^{\alpha-1}\nabla_\theta p_\theta(x)\right]$$

where $\nabla_\theta p_\theta(x)$ is the standard REINFORCE gradient. Xiong et al. (2025b) particularly considers power functions with $\alpha = \frac{1}{2}$, giving the following gradient:

$$\nabla_\theta J_{\text{power-function}} = \mathbb{E}_{x \sim \rho}\left[\frac{1}{2\sqrt{p_\theta(x)}}\nabla_\theta p_\theta(x)\right]$$

In practice, this objective can be optimized by setting the advantage to be $\hat{A}(x,y) = \frac{r(x,y)-\hat{r}}{2\sqrt{\hat{r}}}$ when $\hat{r} > 0$ and 0 otherwise, with $\hat{r}$ being the Monte-Carlo estimate of the pass rate for prompt $x$. This is the variant of the Power Function objective that we will compare MAXRL against in this work.

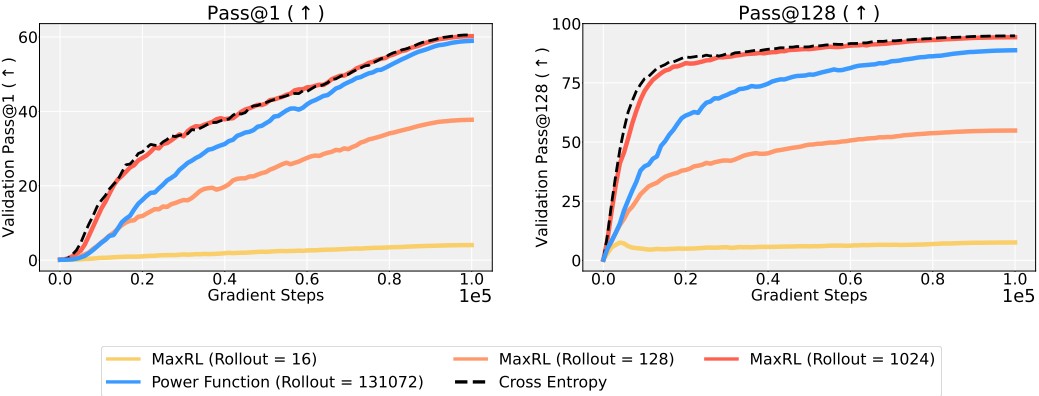

*Figure 24.* (**Comparison with Power Function on supervised ImageNet classification setting**) We compare MAXRL against the Power Function on the ImageNet setting under identical training setup. Despite using significantly more compute, the Power Function remains suboptimal at both pass@1 and pass@128, with the gap being larger at pass@128.

First, we compare against this version of the Power Function on our supervised ImageNet classification setting under an

identical training recipe. Figure 24 shows our empirical findings: MAXRL outperforms Power Function at both pass@1 and pass@128 despite using 128x less rollout compute budget. Moreover, the gap is higher for pass@128, showing that despite performing better than REINFORCE and GRPO in this setting, ultimately the Power Function is a suboptimal choice compared to MAXRL here.

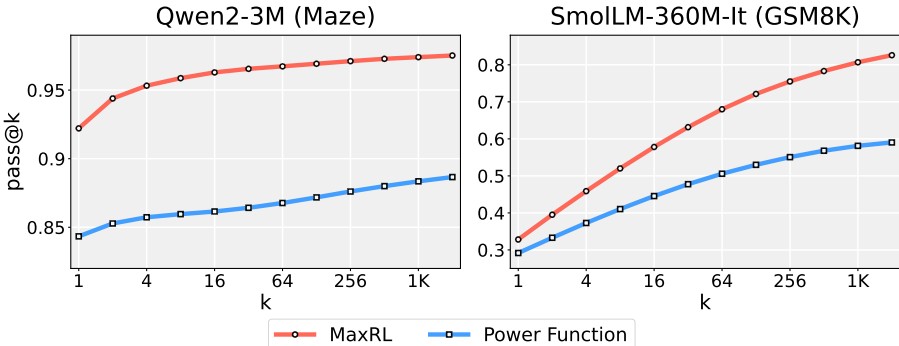

*Figure 25.* (**Comparison with Power Function in the maze and GSM8K setting.**) We compare MAXRL against the Power Function variant mentioned in this section on the infinite data maze and finite data GSM8K training settings under identical training budget and hyperparameters. On both settings, MAXRL significantly outperforms the Power Function at pass@k, for all values of k that we evaluated on.

Next, we scale up our comparisons to the infinite training data maze and finite training data GSM8K settings using identical training hyperparameters and compute budget. Figure 25 summarizes our main findings: MAXRL outperforms this variant of the Power Function objective on both settings. MAXRL achieves significantly higher pass@k on both settings, and outperforms Power Function by $\sim 20\%$ at pass@2048 on GSM8K.

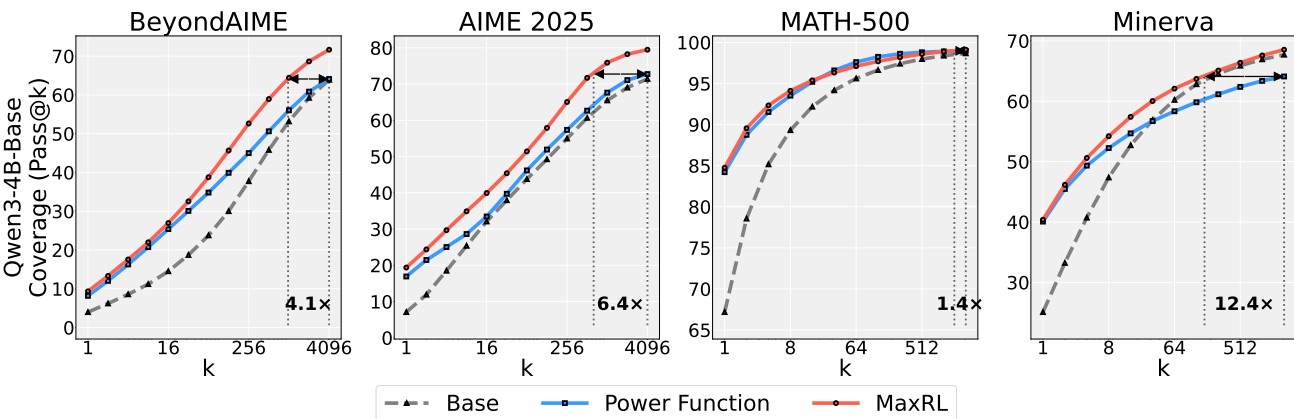

*Figure 26.* (**Comparison with Power Function on Qwen3-4B-Base training**) We compare MAXRL against the Power Function variant mentioned in this section while training a Qwen3-4B-Base model on the POLARIS-53K dataset under identical training setup and compute budget (1000 steps of RL training). MAXRL matches or outperforms the Power Function on evaluation datasets, achieving up to 12.4× gains in test-time scaling efficiency under a perfect verifier.

Finally, we also compared MAXRL against the Power Function on our largest training setting: Qwen3-4B-Base trained on POLARIS-53K. Figure 26 shows our main findings: under identical training setup and compute budget (1000 steps of RL training), MAXRL matches or outperforms Power Function on all 4 evaluation datasets, achieving up to 12.4× test-time scaling efficiency gains under a perfect verifier.

## M. Additional Analyses

### M.1. Non-Zero Gradient Signal Analysis

A prompt for which the model fails to generate any correct rollout contributes no gradient under REINFORCE, GRPO, RLOO, or MAXRL, since the advantage vanishes when every sampled return equals the empirical baseline. The fraction of training prompts with at least one correct rollout therefore serves as a direct measure of how much of the training batch produces nonzero learning signal at each step. We extend the analysis in Figure 7 from the large-scale LLM settings to the maze and SmolLM2-360M-Instruct on GSM8K training settings, and record this fraction throughout training in Figure 27. MAXRL consistently produces at least one correct rollout for a larger fraction of prompts than GRPO and RLOO across both settings, and the gap persists as training progresses. This matches the trends observed in the large-scale experiments and offers a complementary view on why MAXRL continues to improve when REINFORCE-style baselines stall: by upweighting harder prompts and producing larger gradient norms on them (Figure 6), MAXRL keeps a larger fraction of the dataset contributing useful gradient signal even after the easier prompts have saturated.

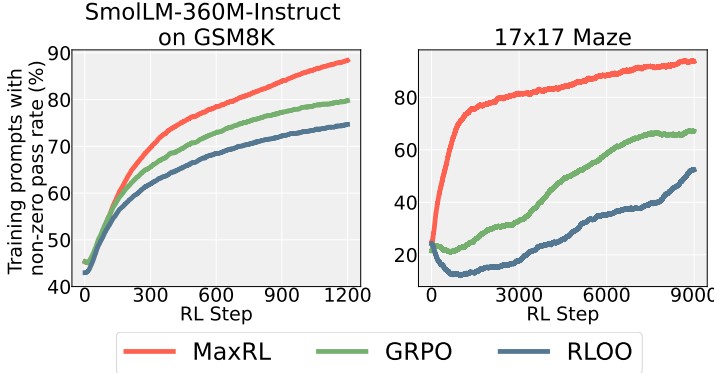

*Figure 27.* (**Fraction of training tasks with non-zero pass rate**) Similar to Figure 6, we also record the fraction of training tasks where the model generates at least one correct rollout on maze and SmolLM2-360M-Instruct training on GSM8K. We see the same trends as Figure 7, and MAXRL consistently outperforms both GRPO and RLOO, demonstrating MAXRL's ability to generate better learning signal during training, as tasks with zero pass-rate contribute no gradients.

### M.2. Policy Gradient Variance Analysis

In this section, we analyze the variance of policy-gradient estimates for MAXRL and other methods, focusing on two questions: (1) does the baseline in our estimator (Equation (10)) actually reduce variance, and (2) how does estimator noise scale with the number of rollouts for MAXRL?

We first study the effect of the control variate in MAXRL. Note that using the unconditional average score function as the baseline is not guaranteed to reduce variance. Previous work such as Greensmith et al. (2004) has noted that this can indeed be suboptimal even for the canonical REINFORCE objective, and provides the derivation of an optimal baseline. However, computing the optimal baseline is complicated in practice, and the average score function as the standard choice of baseline is empirically justified and widely used (Ahmadian et al., 2024; Zeng et al., 2025a). In this section, we follow the same direction and provide empirical proof that the variance reduction technique used by us indeed reduces variance in practical scenarios. Using the rollout pools from the previous experiments, we estimate the trace of the sample covariance matrix across policy-gradient minibatches, following prior work (McCandlish et al., 2018; Zeng et al., 2025a). Given $B$ independently sampled minibatch gradient estimates $\{g_b\}_{b=1}^{B}$, we compute

$$\widehat{\mathcal{V}}(g) = \mathrm{tr}\left(\widehat{\mathrm{Cov}}(g)\right) = \frac{1}{B-1}\sum_{b=1}^{B}\|g_b - \bar{g}\|_2^2, \qquad \bar{g} = \frac{1}{B}\sum_{b=1}^{B} g_b,$$

where each $g_b$ is the flattened policy-gradient vector for one sampled minibatch. Specifically, for GSM8K and Polaris-53K, we use 16 prompts with 512 pre-generated rollouts per prompt; for Maze, we use 256 prompts with 1024 rollouts per prompt. For each $N \in \{4, 8, 16, 32, 64, 128\}$, we randomly subsample $B = 4$ minibatches with $N$ rollouts per prompt and report $\widehat{\mathcal{V}}(g)$. Figure 28 shows that the baseline consistently lowers the variance of the MAXRL gradient estimator across rollout

counts and tasks. The baseline is most important when number of rollouts is low, and eventually for high enough rollout count the gradient variance with and without baseline seems to converge, as expected.

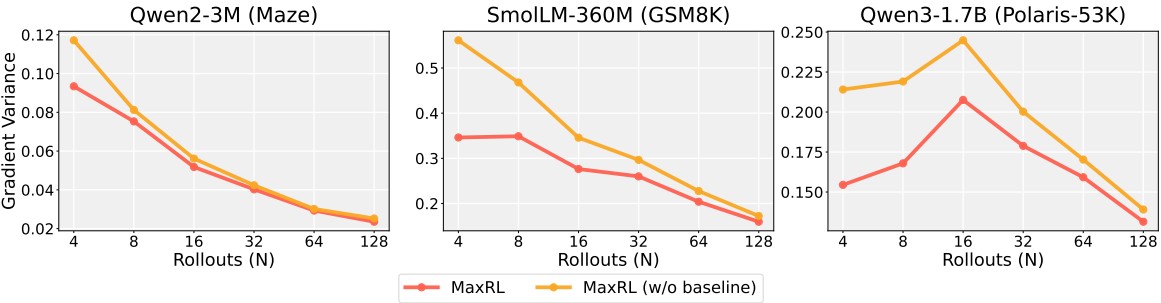

*Figure 28.* (**Effect of the MAXRL baseline on gradient variance**) We estimate the trace of the sample covariance matrix of gradient estimates with and without the proposed baseline for experiments in Section 6. The baseline consistently lowers variance.

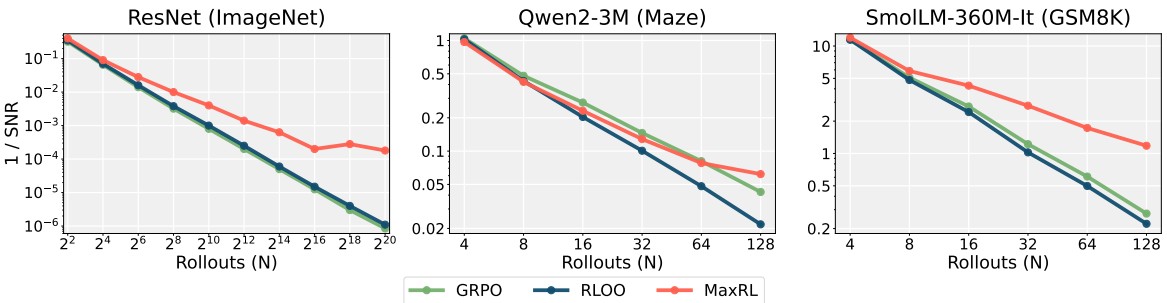

*Figure 29.* (**Inverse gradient SNR across rollout counts**) We compare normalized gradient variance for GRPO, RLOO, and MAXRL. MAXRL becomes increasingly accurate as the number of rollouts grows.

We next study how the signal-to-noise ratio (SNR) of policy gradients scales with the number of rollouts for MAXRL, GRPO, and RLOO. Since these methods optimize different objectives, raw variance is not directly comparable across methods. We therefore normalize by the squared norm of the expected gradient and report inverse signal-to-noise ratio, $\frac{\text{tr}(\text{Cov}(g))}{\|\mathbb{E}[g]\|_2^2}$, where lower values indicate a more accurate gradient estimate relative to its signal. On ImageNet, we estimate this quantity using ResNet-50 gradients over a fixed batch of 1024 images and 64 gradient samples per rollout count. For Maze, we sample 8 prompt batches of 256 prompts from the pre-generated rollout table and estimate inverse SNR from 64 randomized rollout minibatches. For SmolLM on GSM8K, we fix one batch of 64 prompts and form 64 gradient samples through repeated rollout subsampling. Figure 29 shows that the normalized variance of MAXRL decreases as $N$ increases, following the same qualitative trend as GRPO and RLOO.

Together, these results suggest that the proposed baseline reduces the raw variance of the MAXRL estimator. They also show that the stability of MAXRL improves as the rollout budget increases, despite its stronger amplification of very-low-success-rate problems.

### M.3. On the Distribution Sharpening Narrative

Recent work (Yue et al., 2025; Wu et al., 2026) has shown that RL trained models can underperform base models at pass@k, and thereby under a high sampling budget coupled with a perfect verifier, base models can solve a larger number of problems than their RL-trained counterpart. We saw in our experiments that contrary to the standard REINFORCE and GRPO objectives, MAXRL increases pass@k even for very large values of k beyond that of the base model in a majority of scenarios and especially if we do not overtrain. The natural next question to ask is whether MAXRL can lead to expansion of capability beyond the base model. Since any sequence of tokens that lead to the correct answer has a non-zero probability under any language model, we instead ask this question from a finite compute perspective: if one has a large but finite compute, what fraction of the problems can one solve with the base model, and its GRPO- and MAXRL-trained counterparts?

*Table 10.* Per-problem solvability on AIME 2025 with 4096 samples per question. We list the indices of problems solved by each model. Compared to the base model and GRPO-trained model, MAXRL solves three additional problems (**10**, **24**, and **27**) exclusively.

| Model | Solved Problem Indices (AIME 2025) | Solved |
|-------|-------------------------------------|--------|
| Qwen3-4B-Base | 0, 1, 2, 3, 4, 5, 7, 8, 9, 12, 13, 14, 15, 16, 18, 19, 20, 21, 22, 23, 26, 28, 29 | 23/30 |
| GRPO (1000 steps) | 0, 1, 2, 3, 4, 5, 7, 8, 9, 11, 12, 13, 14, 15, 16, 18, 19, 20, 21, 22, 23, 26, 29 | 23/30 |
| MAXRL (1000 steps) | 0, 1, 2, 3, 4, 5, 7, 8, 9, **10**, 11, 12, 13, 14, 15, 16, 18, 19, 20, 21, 22, 23, **24**, 26, **27**, 28, 29 | 27/30 |

Table 10 shows the results of empirical study. Specifically, we take the Qwen3-4B-Base model, and its final GRPO and MAXRL trained checkpoints after 1000 steps of RL training. Next, we generate 4096 rollouts for each of the 30 questions in the AIME 2025 dataset, from each of these 3 models. We list the indices of the problems where each model generates at least 1 correct solution from its 4096 rollouts. We see that the problems solved by the base model and GRPO-trained model are strict subsets of the problem set solved by MAXRL. Moreover, MAXRL solves 3 additional problems exclusively that are not solved by either the base model or the GRPO-trained model.

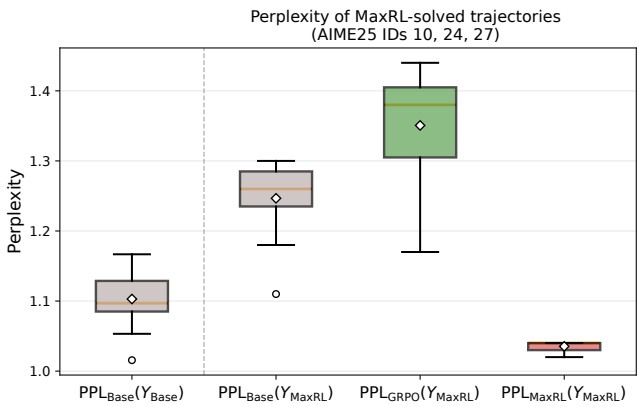

*Figure 30.* (**Perplexity analysis on math problems, using Qwen3-4B-Base and its fine-tuned variants**) We collect $Y_{\mathrm{MaxRL}}$, the 15 correct trajectories from 3 problems on AIME 2025 that only the MAXRL-trained model can solve, and compute the perplexity of each trajectory under all three models. For comparison, we also compute the perplexity of $Y_{\mathrm{Base}}$, an equal number of incorrect rollouts from the base model. The results show that $Y_{\mathrm{MaxRL}}$ have high perplexity under both the base model and the GRPO-trained model, suggesting that MAXRL develops capabilities beyond those present in the base model. Diamonds denote means and circles denote outliers.

Beyond Table 10, we also provide additional evidence in the form of perplexity analysis. Specifically, Figure 30 shows that on the 3 problems from AIME 2025 that only our MAXRL-trained checkpoint could solve, the 15 correct rollouts have very high perplexity under the base and GRPO trained checkpoint. As a control set, we also record perplexity of an equal number of incorrect rollouts generated from the base model, and show that they have much lower perplexity than the MAXRL-generated correct trajectories. Overall, we believe that MAXRL can induce capabilities beyond what is present in the base model, and leave further studying this for future work.

**M.4. Non-Binary Reward Settings**

Our work focuses on binary correctness-based rewards, where the connection to maximum likelihood is most direct. While the binary reward setting already captures a broad and practically important class of RL problems, including applications such as improving complex reasoning in LLMs (e.g., math and code generation) (Zeng et al., 2025b; Mroueh, 2026; Luo et al., 2025) and robotics (Liu et al., 2023; Herzog et al., 2023; Kalashnikov et al., 2018), here we study non-binary reward settings for the sake of completeness.

Interestingly, MAXRL admits a natural extension to non-binary reward settings under mild assumptions. In particular, for non-negative rewards with non-zero expectation under the policy (i.e., $\mathbb{E}_{y \sim \pi_\theta(\cdot|x)}[r(x,y)] > 0$), we can generalize the

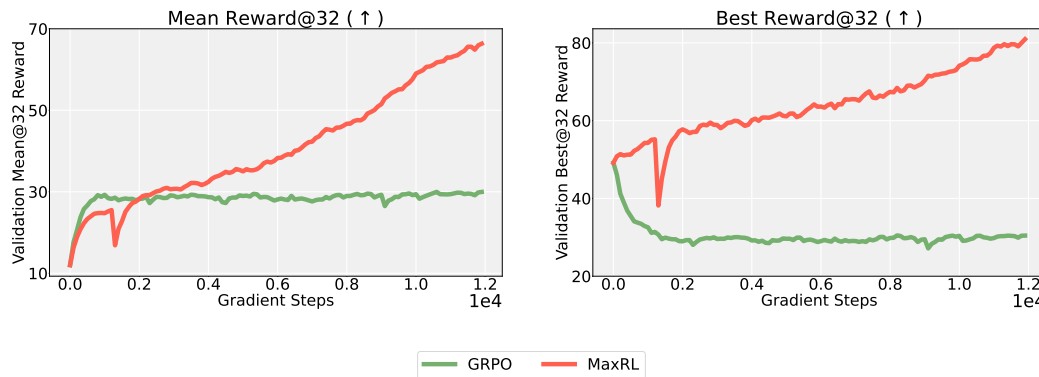

*Figure 31.* (**MAXRL extended to non-binary reward maze setting**) We experiment with our extension of MAXRL in the modified maze setting with non-binary rewards. MAXRL outperforms GRPO in this setting: it achieves 38% higher Mean@32 and 51% higher Best@32 validation reward compared to GRPO. On the other hand, the Best@32 reward for GRPO has collapsed, showing that training for longer is unlikely to improve for GRPO, whereas both Mean and Best reward are still increasing at the end of training for MAXRL.

MAXRL objective from

$$\log \mathbb{E}_{y \sim \pi_\theta(\cdot|x)}[\mathbb{I}[y = y^*]]$$

to

$$\log \mathbb{E}_{y \sim \pi_\theta(\cdot|x)}[r(x, y)]$$

**This yields a gradient estimator analogous to MAXRL**, with advantage:

$$A(x, y_i) = \frac{r(x, y_i) - \hat{\mu}}{\hat{\mu}}$$

where $\hat{\mu}$ is the Monte-Carlo estimate of $\mathbb{E}_{y \sim \pi_\theta(\cdot|x)}[r(x, y)]$.

We have evaluated this extension on a modified version of the Maze task with continuous rewards $r \in [0, 1]$, where shorter trajectories receive higher rewards. Specifically, for maze $x$, let $L_i(x)$ denote the length of the $i$-th rollout, $L_{\min}(x)$ denote the shortest possible path length, and $T$ denote the maximum trajectory length possible (usually equal to the token limit for each generation). We define

$$r_i = \frac{T - L_i(x)}{T - L_{\min}(x)}$$

In this setting, we train a 16M transformer with batch size 256 and 16 rollouts per task, with all other hyperparameters kept the same as the default values for the maze setting. This extension of MAXRL outperforms GRPO by 38% in mean reward and 51% in best@32 reward. Moreover, in this setting, the Best@32 reward for GRPO has collapsed, showing that training for longer is unlikely to improve for GRPO, whereas both Mean and Best reward are still increasing at the end of training for MAXRL.

While the initial results are encouraging, we leave the full theoretical treatment and broader empirical evaluation of non-binary rewards to future work.

