# OpenReview forum: "Maximum Likelihood Reinforcement Learning"
_ICML.cc/2026/Conference — ICML 2026 spotlight_

### Official Review · Reviewer_Ut7h · 2026-03-04

**Soundness:** 3
**Presentation:** 3
**Significance:** 3
**Originality:** 3
**Overall Recommendation:** 5
**Confidence:** 3

**Summary:**

This paper introduces a new framework called MaxRL, which rethinks the RL objective in correctness-based tasks with non-differentiable generation. The authors argue that in such settings, standard RL algorithms are misaligned with the principle of maximum likelihood (ML) training. The gradient of the RL objective weights all inputs roughly uniformly, causing optimization to be dominated by easy examples. In contrast, the gradient of the ML objective reweights by the inverse success probability, concentrating effort on failing or uncertain cases. The paper states that the ML objective admits a Maclaurin series expansion in terms of pass@k probabilities, and that standard RL optimizes only the first term. MaxRL is defined by truncating this series at a finite order T, and intermediate T values provide a smooth interpolation between standard RL and exact ML.

**Compliance With Llm Reviewing Policy:**

Affirmed.

**Key Questions For Authors:**

See above.

**Limitations:**

See above.

**Strengths And Weaknesses:**

**Strengths**

1. This paper identifies the intrinsic connection between traditional RL objectives and the maximum likelihood objective, which explains phenomena like distribution sharpening and pass@k degradation observed in modern RL.
2. This paper provides a key insight that establishes the relationship between RL and ML through a Maclaurin expansion, demonstrating that RL is a first-order approximation of the ML objective in correctness-based tasks.
3. Theorem 4.2 derives a practically simple and unbiased gradient estimator for the MaxRL objective in Equation 5.
4. Comprehensive experiments on non-differentiable correctness-based tasks validate the effectiveness of the proposed MaxRL framework.

**Weaknesses**

My only concern is that this paper mainly focuses on the correctness-based scenario. However, a more generalized case would be applying this method to tasks beyond binary classification signals, which limits the applicability of MaxRL.

---

> ### Author Rebuttal · Authors · 2026-03-31
>
> We thank the reviewer for the positive assessment and helpful suggestion.
>
> > My only concern is that this paper mainly focuses on the correctness-based scenario...
>
> ## Extension to non-binary rewards
>
> Our work focuses on binary correctness-based rewards, where the connection to maximum likelihood is most direct. However, **MaxRL admits a natural extension to non-binary reward** settings under mild assumptions. In particular, for non-negative rewards with non-zero expectation under the policy (i.e., $E_{y \sim \pi_\theta(.|x)}[r(x, y)] > 0$), we can **generalize the objective** from
>
> $$ \log E_{y \sim \pi_\theta(.|x)}[I[y = y^*]]$$
>
> to
>
> $\log E_{y \sim \pi_\theta(.|x)}[r(x, y)]$
>
> **This yields a gradient estimator analogous to MaxRL**, with advantage
>
>
> $$A(x, y_i) = \frac{r(x, y_i) - \hat{\mu}}{\hat{\mu}}$$
>
> where $\hat{\mu}$ is the Monte Carlo estimate of $E[r(x,y)]$.
>
> ## Preliminary empirical results
>
> **We have evaluated this extension on a modified version of the Maze task** (Section 6.2) with continuous rewards $r \in [0,1]$, where shorter trajectories receive higher rewards. Specifically, for maze $x$, let $L_i(x)$ denote the length of the $i$-th rollout, $L_{\min}(x)$ denote the shortest possible path length, and $T$ denote the maximum trajectory length. We define
>
> $$r_i = \frac{T - L_i(x)}{T - L_{\min}(x)}$$
>
> In this setting, we train a 16M transformer with batch size 256 and 16 rollouts per task. **This extension of MaxRL outperforms GRPO by 38% in mean reward and 51% in best@32 reward**. [See this figure](https://ibb.co/9H6bQTwP) for complete results.
>
> While the initial results are encouraging, we leave the full theoretical treatment and broader empirical evaluation of non-binary rewards to future work. Nonetheless, we emphasize that **the binary reward setting already captures a broad and practically important class of RL problems**, including applications such as improving complex reasoning in LLMs (e.g., math and code generation) [1,2,3,4] and robotics [5,6,7].
>
> ## Additional experiments
>
> We have run additional experiments to compare MaxRL against different baselines following the suggestions by the other reviewers. Here we mention a few of them to showcase the general efficacy of MaxRL:
>
> 1. We have included comparison against the Power Function baseline from Xiong et al. [8], as a potential alternative for optimizing non-linear functions of the pass rate. Across all settings ([Qwen3-4B-Base](https://ibb.co/cSWbN224), [Maze](https://ibb.co/jvQJXh5F), [ImageNet](https://ibb.co/ymJSVs0k ), [GSM8K](https://ibb.co/DDcWhjfc)), MaxRL outperforms the power function baseline, **achieving up to 12.4x test-time scaling efficiency gains on Qwen3-4B-Base**.
>
> 2. We have added a direct comparison between MaxRL and curriculum learning based algorithm ZPD, which reweights per-prompt gradients as $w(p) = p(1 - p)$. Across all evaluated settings, MaxRL consistently outperforms ZPD in both pass@1 and pass@k. On [Qwen3-4B-Base](https://ibb.co/S1TKLBQ) and under identical training conditions, MaxRL outperforms ZPD at all pass@k, achieving up to **29.0× test-time scaling efficiency gains**.
>
> **We hope we have addressed all of the reviewer’s concerns. We would love to engage with the reviewer and respond to any other enquiries they may have.**
>
> ## References
>
> [1] SimpleRL-Zoo: Investigating and Taming Zero Reinforcement Learning for Open Base Models in the Wild
>
> [2] DeepSeek-R1: Incentivizing Reasoning Capability in LLMs via Reinforcement Learning
>
> [3] Reinforcement Learning with Verifiable Rewards: GRPO's Effective Loss, Dynamics, and Success Amplification
>
> [4] DeepCoder: A Fully Open-Source 14B Coder at O3-mini Level
>
> [5] LIBERO: Benchmarking Knowledge Transfer for Lifelong Robot Learning
>
> [6] Deep RL at Scale: Sorting Waste in Office Buildings with a Fleet of Mobile Manipulators
>
> [7] QT-Opt: Scalable Deep Reinforcement Learning for Vision-Based Robotic Manipulation
>
> [8] Reinforce-Ada: An Adaptive Sampling Framework under Non-linear RL Objectives

---

> > ### Author Rebuttal · Reviewer_Ut7h · 2026-04-03
> >
> > My concerns have been adequately addressed. I wiil keep my score.

---

> > > ### Author Response · Authors · 2026-04-03
> > >
> > > Dear Reviewer Ut7h,
> > >
> > > Thanks for your positive assessment of our work and helpful feedback! **We are glad to know that we have adequately addressed your concerns**.
> > >
> > > Thanks again,
> > >
> > > Authors

---

### Official Review · Reviewer_XQZj · 2026-03-08

**Soundness:** 3
**Presentation:** 3
**Significance:** 3
**Originality:** 3
**Overall Recommendation:** 5
**Confidence:** 3

**Summary:**

This paper addresses the mismatch between the maximum likelihood (ML) objective and the expected-reward objective used in reinforcement learning (RL) for correctness-based problems with non-differentiable sampling. The authors show that for binary reward tasks, the standard RL objective (expected reward) is merely a first-order Taylor approximation of the log-likelihood, discarding higher-order curvature information that upweights hard, low-pass-rate inputs. They introduce Maximum Likelihood Reinforcement Learning (MaxRL), a compute-indexed family of objectives obtained by truncating a Maclaurin (failure-series) expansion of the log-likelihood at order T, which interpolates between standard RL ($T=1$) and exact ML ($T \to \infty$).

**Compliance With Llm Reviewing Policy:**

Affirmed.

**Key Questions For Authors:**

1. **On the distribution sharpening narrative:** You mention recent work (Yue et al., 2025; Wu et al., 2026) suggests that RLVR's sharpening may partly reflect the model exploiting already-learned behaviors rather than genuine mode collapse (Line 36). Does MaxRL actually teach new capabilities, or does it simply sharpen more slowly? Do you have any experiments to quantify this?
2. **On the conditional estimator's K=0 behavior:** The estimator sets the gradient to zero when all K samples fail. But the ML gradient at those inputs is *largest* (weight 1/p). Isn't there a fundamental tension between what the population objective demands and what the estimator provides? Could importance-weighted or rejection-sampling approaches alleviate this?
3. **On the failure-series convergence:** First, there seems to be a typo in Eq. 2, $\nabla J_{ml}$ should be $J_{ml}$. Second, the Maclaurin expansion converges for p in (0, 1], but the gradient sum (Eq. 2) involves $(1-p)^k / k$, which converges slowly for small p. For a prompt with p=0.01 and T=128, how much of the ML gradient is captured? Does the slow convergence for hard prompts undermine the motivation for finite-T truncation? Can you develop a non-asymptotic analysis for the ML gradient based on $p$ and $N$?
4. **Comparison with Xiong et al. (2025b):** You mention this concurrent work investigates non-linear functions of pass rate. How do their proposed weighting functions compare with your $w_T(p)$? Is there empirical evidence that the specific ML-derived weighting is superior to other non-linear reweightings, for instance power function in Xiong et al. (2025b)?

**Limitations:**

yes

**Strengths And Weaknesses:**

# Strength

1. **Elegant and clarifying theoretical framework.** The Maclaurin expansion of the ML objective into pass@k gradients (Eq. 2-3) is a clean, insightful result that reframes the RL vs ML distinction in a precise, quantitative way. The weight-function view (Table 2, Figure 1) is particularly illuminating, showing that RL, GRPO, MaxRL, and ML form a natural spectrum of objectives differing only in how they weight inputs by difficulty. This framework has explanatory value beyond the specific algorithm proposed.
2. **Minimal algorithmic change with strong theoretical backing.** The practical algorithm amounts to a single change: normalize by K (number of successes) instead of N (total samples). This simplicity is a strength, as argued by the author; it means the method is trivially implementable in any existing GRPO/REINFORCE codebase. The fact that this small change targets a fundamentally different population-level objective (Theorem 4.2) is surprising and well-proven.
3. **Comprehensive and well-designed experiments.** The experimental evaluation spans four domains of increasing complexity: (1) ImageNet as a controlled didactic setting where exact ML is available, (2) procedural mazes for infinite-data RL, (3) GSM8K for data-scarce RL, and (4) Qwen3 models (1.7B, 4B) on competition math for large-scale validation. Each domain serves a distinct purpose, and the progression is well-motivated. The inclusion of multiple baselines (GRPO, RLOO, PKPO, Differential Smoothing, SELF) is thorough.

# Weakness
1. **Binary reward assumption is restrictive.** The entire framework is built on binary (success/failure) rewards. The paper acknowledges this limitation (Section 8) but does not discuss how the approach might extend to continuous or multi-level rewards. Many practical RL settings involve partial credit, process rewards, or continuous reward signals. Without even a sketch of how MaxRL generalizes, the scope of the contribution is limited to a specific (like RL alignment, though important) subclass of problems.
2. **The role of T is underdeveloped.** The truncation level T is the key hyperparameter that controls the compute-accuracy tradeoff, yet its treatment is surprisingly thin. The paper shows that $T=1$ recovers Reinforce and $T=\infty$ recovers MaxRL with the conditional estimator in Figure 1, but there is no guidance on how to choose T in practice. In the experiments, the practical algorithm (Algorithm 1) uses the conditional estimator (equivalent to T=N), so the interpolation between RL and ML, is never actually explored empirically at intermediate T values. This undermines the "compute-indexed family" narrative.
3. **Variance analysis is incomplete.**  The paper acknowledges that the estimator (Eq. 6) has high variance when K is small (Section 4.1) and proposes a simple control variate (Eq. 7) (Section 4.2). However, there is no formal variance analysis comparing the MaxRL estimator to REINFORCE or GRPO estimators. Given that the estimator normalizes by K (which can be 0 or 1), understanding the variance behavior as a function of pass rate and sample size is critical. The K=0 case (all failures) produces zero gradient, which means MaxRL gets no learning signal at all on the hardest prompts, precisely the ones ML would weight most heavily. This tension between the population-level objective (which upweights hard inputs) and the estimator (which gets zero signal on the hardest inputs) deserves more analysis.
4. **The connection to exact ML is only asymptotic.** The paper shows that MaxRL approaches ML as T (or N) -> $\infty$, but in practice, the number of rollouts is finite (16-128 in the LLM experiments). For prompts with very low pass rates (e.g., p = 0.01 with N = 16), the estimator is almost always zero. The paper does not quantify how far the practical estimator is from the true ML gradient for realistic rollout budgets. The author is suggested to provide a non-asymptotic analysis (theoretical or empirical) of the approximation gap as a function of N and p, showing under what conditions the practical estimator is a good approximation to ML.

---

> ### Author Rebuttal · Authors · 2026-03-31
>
> We thank the reviewer for their thoughtful review and suggestions.
>
> > Binary reward assumption is restrictive
>
> **A natural extension to continuous rewards is to optimize $\log E[r]$**, where $r \in [0,1]$ is a normalized reward. This yields an estimator analogous to MaxRL in the non-binary setting: $$A(x, y_i) = \frac{r(x, y_i) - \mu}{\mu}$$ where $\mu$ is a Monte Carlo estimate of $E[r(x,y)]$. **In preliminary experiments on a continuous reward variant of Maze**, **this approach outperforms GRPO** ([results](https://ibb.co/9H6bQTwP)).
>
> > so the interpolation between RL and ML…
>
> In our framework, $T$ is not an independent hyperparameter. By Theorem 2, our estimator with $N$ rollouts is **exactly** unbiased for the truncated objective with $T = N$. Thus, the truncation level is implicitly determined by the compute budget (number of rollouts). Empirically, we refer to Figure 2 and 3 to show the effect of increasing T in the ImageNet and Maze settings, respectively: increasing $T$ results in stronger performance. We have also run additional experiments on the GSM8K setting ([Figure](https://ibb.co/j9D8cvtC)) which shows similar behavior. Therefore, one would ideally choose the highest T (= N) that they can afford within their compute budget.
>
> > Variance analysis is incomplete
>
> We have conducted an empirical study comparing the variance of the MaxRL, GRPO, and RLOO estimators as a function of the number of samples. Since these estimators correspond to different objectives, we normalize the variance by the squared norm of the expected gradient and report the resulting normalized variance (inverse signal-to-noise ratio) on the ImageNet setting. MaxRL exhibits a variance decay trend (like RLOO/GRPO), indicating that its estimator becomes increasingly more accurate as the sample size increases despite targeting a harder objective ([see this Figure as an analysis of variance as a function of N](https://ibb.co/tN9L1Qt)). Variance also depends on the pass rate $p$: we ran an analysis on ImageNet where we fixed N to be 128 and measured variance vs signal. [This Figure](https://ibb.co/hRmz6H6Q) shows our result: as expected, the variance of the estimator grows as $p$ goes to 0, requiring larger $N$ for these harder tasks. We leave a full theoretical analysis of variance for future work.
>
> We thank the reviewer for pointing out the critical case when K = 0. **Gradient signal requires observing at least one success**. With per-sample success probability $p$, this occurs with probability $1 - (1-p)^N$, implying that $O(1/p)$ samples are needed in expectation to obtain signal. Thus, in the low-p regime, **zero-signal updates are unavoidable unless the number of samples is increased**.
>
> As the reviewer suggests, adaptive allocation directly addresses this. In contrast, importance weighting or rejection sampling cannot recover signal when $K=0$, as no successful samples are observed.
>
> > The connection to exact ML is only asymptotic.
>
> **We can in fact provide a precise, non-asymptotic characterization of the approximation gap**. The population-level gradient of MaxRL at order T uses weight $w_T(p) = \frac{1 − (1−p)^T}{p}$, while exact ML uses $w_{ML}(p) = \frac{1}{p}$. The ratio of these weights gives the *fraction of the ML gradient signal* captured by MaxRL at order T:
>
> $$w_T(p) / w_{ML}(p) = 1 − (1−p)^T = pass@T$$
>
> This is a clean, interpretable result: the fraction of the ML gradient recovered by MaxRL on a prompt with pass rate $p$ is exactly the pass@T probability for that prompt. This holds for all finite T and p ∈ (0, 1], with no asymptotics involved. Please see [this Figure](https://ibb.co/j9VHfJcs) for non-asymptotic approximation gap as a function of $N (= T)$ and $p$, and [this Figure](https://ibb.co/ZRY2zyJm) for a heatmap of the convergence rate, specifically showing the approximation error for $p = 0.01$ and $T = 16$.
>
> > On the distribution sharpening narrative
>
> On Qwen3-4B-Base, we generate 4096 rollouts per prompt on AIME 2025 and identify problems with at least one correct solution. MaxRL solves 26/30 problems, compared to 23/30 for the base model ([results](https://ibb.co/DPqc5Sc8)). To assess whether these are simply reweighted existing behaviors, we compute the perplexity of correct solutions under the base model. For the additional problems solved by MaxRL, the corresponding solutions have very high perplexity under the base model, indicating that they lie in low-probability regions of the original distribution, see [this Figure](https://ibb.co/7f4XryC).
>
> > Comparison with Xiong et al. (2025b)
>
> Across all settings ([Qwen3-4B-Base](https://ibb.co/cSWbN224), [Maze](https://ibb.co/jvQJXh5F), [ImageNet](https://ibb.co/ymJSVs0k ), [GSM8K](https://ibb.co/DDcWhjfc)), MaxRL outperforms the power function baseline, achieving up to 12.4x test-time scaling efficiency gains on Qwen3-4B-Base. We will add this to our paper, and leave studying other choices for non-linear objectives of the pass rate for future work.

---

> > ### Author Rebuttal · Reviewer_XQZj · 2026-04-02
> >
> > I acknowledge the effort the authors have put into this rebuttal, and my concerns are addressed. I will maintain my score.

---

> > > ### Author Response · Authors · 2026-04-02
> > >
> > > Dear Reviewer XQZj,
> > >
> > > Thanks a lot for your thoughtful questions and suggestions, they would help improve our paper significantly! We are also grateful for your positive review of our work.
> > >
> > > Thanks,
> > >
> > > Authors

---

### Official Review · Reviewer_eqnB · 2026-03-18

**Soundness:** 3
**Presentation:** 4
**Significance:** 3
**Originality:** 3
**Overall Recommendation:** 5
**Confidence:** 5

**Summary:**

This paper focuses on binary correctness tasks with non-differentiable sampling. The authors first show that the standard reinforcement learning objective (expected reward maximization) can be interpreted as a first-order approximation of the maximum likelihood objective.

Building on this insight, they propose a simple, unbiased gradient estimator for the maximum likelihood objective. To account for computational constraints, they introduce MaxRL, a method that progressively improves the quality of the maximum likelihood approximation as computation increases.

Empirically, the paper evaluates MaxRL against several baselines, including RLOO, GRPO, and pass@k optimization methods, across tasks such as binary classification, maze navigation, and LLM-based mathematical reasoning. The results demonstrate consistent improvements over these baselines.

**Compliance With Llm Reviewing Policy:**

Affirmed.

**Final Justification:**

The authors have addressed my key questions (please see strengths and weaknesses). Therefore, I am increasing my score by 1.

**Key Questions For Authors:**

[1] Relation to curriculum-based RL (ZPD weighting)

RL(OO) with a "not too easy, not too hard" (Zone of Proximal Development, ZPD) curriculum (Bae et al., 2025) can be interpreted as reweighting prompts with $w(p) = p (1-p)$ (as in Table 2).

It would be interesting to include this variant as a baseline in the experiments (e.g., Figures 3-5), to better understand how MaxRL compares to difficulty-aware sampling strategies.

Reference: Bae et al., 2025. Online Difficulty Filtering for Reasoning-Oriented Reinforcement Learning.

[2] pass@k optimization details and evaluation alignment

In Tables 3 and 4, the pass@k optimization method (Walder & Karkhanis, 2025) is trained with $k=16$ but evaluated across multiple $k$ values. If the evaluation metric is known in advance, would training with the same target $k$ yield better performance?

Could the authors include a pass@k optimization baseline in Figure 5 for completeness?

In Table 2, what is the corresponding weighting function $w(p)$ for pass@k optimization?

[3] GRPO behavior in a supervised learning setting

In the supervised learning experiment (Figure 2), how does GRPO perform?
Would it behave similarly to REINFORCE in this setting?

[4] Variance reduction analysis

It is not immediately clear how the gradient estimator in Eq. (7) reduces the variance compared to Eq. (6). Could the authors provide an analytical comparison of the variance of the two estimators?

[5] Beyond binary rewards

Do the authors have insights on extending the proposed method or analysis to non-binary or continuous reward settings?

Minor Comments

There is a typo in Eq. (2): $\nabla_\theta$ appears to be unnecessary.

**Limitations:**

yes

**Strengths And Weaknesses:**

**Strengths**

Technical soundness and clarity: The paper is well-written, easy to follow, and clearly structured.

Important problem setting: Binary correctness tasks with non-differentiable objectives are highly relevant, especially in reasoning settings.

Novel and principled contribution: The connection between RL objectives and maximum likelihood is insightful, and the proposed estimator is both principled and elegant.

Practical advantages:
- The method is simple and easy to implement.
- It provides a compute-performance trade-off, where increasing compute improves the approximation quality.

Strong empirical validation: The method consistently outperforms competitive baselines (RLOO, GRPO, pass@k optimization) across multiple domains.

**Weaknesses / Limitations**

Limited scope of analysis: The method and theoretical analysis are restricted to binary reward settings, and it is unclear how they generalize to more complex or continuous reward scenarios.

Comparative coverage: Some relevant baselines and perspectives (e.g., curriculum-based weighting approaches) are not explored.

---

> ### Author Rebuttal · Authors · 2026-03-31
>
> We thank the reviewer for their valuable feedback. We address their concerns below:
>
> > Zone of Proximal Development
>
> We have **added a direct comparison between MaxRL and the reviewer-suggested ZPD variant**, which reweights per-prompt gradients as $w(p) = p(1 - p)$. Across all evaluated settings, **MaxRL consistently outperforms ZPD**.
>
> We will include detailed comparisons in the revised paper; key results are summarized below:
> - [ImageNet](https://ibb.co/NgmDMgcb): ZPD behaves similarly to REINFORCE and does not yield measurable training improvements.
> - [Maze](https://ibb.co/r2jCx835): MaxRL outperforms ZPD across all pass@k values.
> - [GSM8K](https://ibb.co/hRvXhpjJ): MaxRL achieves larger gains, particularly at higher pass@k.
> - [Qwen3-4B-Base](https://ibb.co/S1TKLBQ): Under identical training conditions, MaxRL outperforms ZPD at all pass@k, achieving up to **29.0× test-time scaling efficiency gains**.
>
> We will incorporate these results into the final version of our paper.
>
> > If the evaluation metric is known in advance...
>
> We thank the reviewer for this insightful question. We have **added an evaluation of pass@16 comparing PKPO (with $K=16$) and MaxRL**, ensuring that PKPO is evaluated under the same setting it optimizes during training. We **find that MaxRL substantially outperforms PKPO on pass@16**, despite PKPO directly optimizing this metric. A similar phenomenon is observed with REINFORCE, which directly optimizes pass@1 yet consistently underperforms MaxRL at pass@1. **This suggests that directly optimizing a specific pass@k objective does not necessarily yield the best performance at that $k$**. We hypothesize that MaxRL’s aggregation of signals across pass@k events enables more effective use of the learning signal, and we leave a detailed investigation to future work.
>
> We report pass@16 results for PKPO with $K=16$ below:
>
> **(Maze)**
>
> | Method | Pass@1 (%) | Pass@16 (%) | Pass@128 (%) |
> |---------------|------------|-------------|--------------|
> | PKPO (K=16) | 74.49 | 76.68 | 77.65 |
> | MaxRL | **84.39** | **92.16** | **93.96** |
>
> **(GSM8K)**
>
> | Method        | Pass@1 | Pass@16 | Pass@128 |
> |---------------|--------|---------|----------|
> | PKPO (K=16)   | 30.72  | 53.56   | 67.24    |
> | MaxRL         | **33.21**  | **60.29**   | **74.98**    |
>
> We will include and discuss this result in the final version.
>
> > Could the authors include a pass@k optimization baseline...
>
> Due to time constraints, we ran the PKPO baselines only for Qwen3-4B-Base. We use batch size 256 and N = 16, then train with PKPO for $K \in \{4, 8, 16\}$. Consistent with Maze and GSM8K, MaxRL matches or outperforms PKPO with $K \in \{4, 8, 16\}$ across all four test datasets, with MaxRL significantly outperforming PKPO on harder datasets such as BeyondAIME and AIME 2025 at all pass@k, and achieving similar performance on MATH-500 and Minerva. Full results are available [here](https://ibb.co/Kj5Jt80L) and will be included in the final version.
>
> > In Table 2, what is the corresponding weighting function  for pass@k optimization?
>
> The weight function $w(p)$ for PKPO with $K = k$ is:
>
> $$w(p) = k(1 - p)^{k - 1}$$
>
> > GRPO behavior in a supervised learning setting
>
> We refer the reviewer to **Figure 9 in the Appendix**, which reports GRPO’s performance in the supervised ImageNet classification setting. We will add a pointer to this result in the main paper in the next revision; for convenience, the comparison figure is available [here](https://ibb.co/t0D5Hx1).
>
> In this setting, GRPO substantially outperforms REINFORCE due to its $\frac{1}{\sqrt{p(1 - p)}}$ gradient weighting, which emphasizes harder examples (cf. Figure 1). However, **GRPO still underperforms MaxRL at both pass@1 and pass@k**, indicating that its weighting scheme remains suboptimal.
>
> > Beyond binary rewards
>
> This is an interesting direction that we are actively exploring. One possible explanation for MaxRL’s strong performance is that the log transformation effectively convexifies the objective, leading to improved training dynamics. Motivated by this, **a natural extension to continuous rewards is to optimize $\log E[r]$**, where $r \in [0,1]$ is a normalized reward. This yields an estimator analogous to MaxRL in the non-binary setting, namely $$A(x, y_i) = \frac{r(x, y_i) - \mu}{\mu}$$ where $\mu$ is a Monte Carlo estimate of $E[r(x,y)]$. .
>
> **In preliminary experiments on a continuous variant of Maze**, where rewards are proportional to the (negative) time to exit, **this approach outperforms GRPO** ([results](https://ibb.co/9H6bQTwP)) by 38% and 51% at mean@32 and best@32 validation reward, respectively. While these results are encouraging, we leave a full theoretical treatment to future work.
>
> > Minor Comments
>
> Thanks for catching the typo: we will fix it in the next version of the paper!
>
> **We hope our rebuttal addresses all of the reviewer’s concerns. Otherwise, we would love to engage with the reviewer to address any of their remaining concerns.**

---

> > ### Author Rebuttal · Reviewer_eqnB · 2026-04-02
> >
> > The authors have addressed my key questions. I will increase my score.

---

> > > ### Author Response · Authors · 2026-04-02
> > >
> > > Dear Reviewer eqnB,
> > >
> > > We are glad that we have addressed your concerns! We again thank you for your thoughtful experiment suggestions and feedback, they will help us greatly in improving our work.
> > >
> > > Thanks,
> > >
> > > Authors

---

### Decision · Program_Chairs · 2026-04-30

**Decision:**

Accept (spotlight)

**Comment:**

This paper introduces MaxRL, a method that bridges the gap between reinforcement learning and maximum likelihood for tasks with clear right or wrong answers. All reviewers strongly praised the work for its elegant math, incredibly simple implementation, and impressive performance improvements on complex tasks like math reasoning. The authors did an excellent job addressing all questions during the rebuttal—including showing how the method can work for non-binary rewards—making this a top-quality paper that is highly recommended for an oral presentation.